# Diphthamide deficiency promotes association of eEF2 with p53 to induce p21 expression and neural crest defects

Yu Shi [1,2,9] ✉, Daochao Huang [3,9], Cui Song [4,9], Ruixue Cao [5], Zhao Wang [5], Dan Wang [3], Li Zhao [3], Xiaolu Xu [2], Congyu Lu [2], Feng Xiong [4], Haowen Zhao [1,3], Shuxiang Li [3,4], Quansheng Zhou [3,4], Shuyue Luo [3], Dongjie Hu [3], Yun Zhang [6], Cui Wang [6], Yiping Shen [7], Weiting Su [8], Yili Wu [5], Karl Schmitz [2], Shuo Wei [2,10] ✉ & Weihong Song [5,10] ✉

Diphthamide is a modified histidine residue unique for eukaryotic translation elongation factor 2 (eEF2), a key ribosomal protein. Loss of this evolutionarily conserved modification causes developmental defects through unknown mechanisms. In a patient with compound heterozygous mutations in *Diphthamide Biosynthesis 1* (*DPH1*) and impaired eEF2 diphthamide modification, we observe multiple defects in neural crest (NC)-derived tissues. Knockin mice harboring the patient's mutations and *Xenopus* embryos with Dph1 depleted also display NC defects, which can be attributed to reduced proliferation in the neuroepithelium. DPH1 depletion facilitates dissociation of eEF2 from ribosomes and association with p53 to promote transcription of the cell cycle inhibitor p21, resulting in inhibited proliferation. Knockout of one *p21* allele rescues the NC phenotypes in the knockin mice carrying the patient's mutations. These findings uncover an unexpected role for eEF2 as a transcriptional coactivator for p53 to induce p21 expression and NC defects, which is regulated by diphthamide modification.

Eukaryotic elongation factor 2 (eEF2) is necessary for ribosome translocation over the bound mRNA during translation elongation. It is highly conserved throughout evolution, and all archaeal and eukaryotic eEF2 orthologs contain a modified histidine residue called diphthamide, which has not been found in any other proteins[1,2]. Diphthamide modification of eEF2 is catalyzed by diphthamide biosynthesis (DPH) enzymes, with the initial cleavage of S-adenosylmethionine (SAM) to generate a 3-amino-3-carboxylpropyl

---

[1]Department of Clinical Laboratory, Children's Hospital of Chongqing Medical University, Chongqing Key Laboratory of Child Neurodevelopment and Cognitive Disorders, Ministry of Education Key Laboratory of Child Development and Disorders, National Clinical Research Center for Child Health and Disorders, 136 Zhongshan 2nd Rd, Chongqing 400014, China. [2]Department of Biological Sciences, University of Delaware, Newark, DE 19716, USA. [3]Department of Pediatric Research Institute, Children's Hospital of Chongqing Medical University, 136 Zhongshan 2nd Rd, Chongqing 400014, China. [4]Department of Endocrinology and Genetic Metabolism Disease, Children's Hospital of Chongqing Medical University, 136 Zhongshan 2nd Rd, Chongqing 400014, China. [5]Oujiang Laboratory (Zhejiang Lab for Regenerative Medicine, Vision and Brain Health), Institute of Aging, Key Laboratory of Alzheimer's Disease of Zhejiang Province, School of Mental Health and Kangning Hospital, The Second Affiliated Hospital and Yuying Children's Hospital, Wenzhou Medical University, Wenzhou 325035 Zhejiang, China. [6]Department of Radiology, Children's Hospital of Chongqing Medical University, 136 Zhongshan 2nd Rd, Chongqing 400014, China. [7]Division of Genetics and Genomics, Boston Children's Hospital and Harvard Medical School, Boston, MA 02115, USA. [8]Kunming Institute of Zoology, Chinese Academy of Science, Kunming 650223 Yunnan, China. [9]These authors contributed equally: Yu Shi, Daochao Huang, Cui Song. [10]These authors jointly supervised this work: Shuo Wei, Weihong Song. ✉e-mail: shiyu@hospital.cqmu.edu.cn; swei@udel.edu; weihong@wmu.edu.cn

---

(ACP) radical and subsequent transfer of the ACP group to the histidine residue on eEF2 carried out by a Dph2 homodimer in archaea or a Dph1-2 heterodimer in eukaryotes. The ACP-histidine intermediate then undergoes further methylation and amidation to yield the final diphthamide product (Supplementary Fig. 1)[1–4]. Several bacterial toxins, including the lethal diphtheria toxin, can transfer ADP ribose specifically to the diphthamide on eEF2, causing eEF2 inactivation and cell death[1,2]. However, no endogenous ADP-ribosyltransferase has been identified to target diphthamide[2].

Autosomal recessive mutations in human *DPH1* or *DPH2* have been linked to developmental delay with short stature, dysmorphic features, and sparse hair (DEDSSH1 and DEDSSH2; OMIM 616901 and 620062, respectively), rare genetic syndromes with characteristic craniofacial disorders, growth retardation, and intellectual disability[4–8]. The symptoms are reminiscence of the phenotypes of *DPH1* knockout (KO) mice, which are largely rescued by replacing the endogenous eEF2 with a G717R mutant that mimics diphthamide-modified eEF2[1,9,10], suggesting that they are caused by disrupted eEF2 diphthamide modification. *Dph1*$^{-/-}$ mouse embryonic fibroblast cells have reduced proliferation[1,9], which may explain the smaller size of the KO mice. However, diphthamide deficiency does not seem to affect the growth rate of the yeast *S. cerevisiae* or human MCF7 cells[2,11]. At the molecular level, diphthamide deficiency increases −1 frameshift in translation elongation, but this effect does not seem to be the cause of the morphological anomalies displayed by *DPH1*-KO mice[1]. Thus, the underpinning mechanisms for diphthamide in embryonic development and DEDSSH pathogenesis remain elusive.

Here we describe a DEDSSH1 patient with compound heterozygous *DPH1* mutations that caused diphthamide deficiency and developmental defects in neural crest (NC)-derived tissues. We found that diphthamide deficiency promotes the association of eEF2 with p53, leading to enhanced binding of p53 to *p21* promoter to induce the expression of more p21, which suppresses cell proliferation in the neuroepithelium including the NC. These results provide new insights into the pathophysiology of birth defects caused by diphthamide deficiency in DEDSSH patients.

## Results

### A DEDSSH patient with compound heterozygous mutations in *DPH1*

We have identified a four-year old girl with global developmental delay and intellectual disability (see Supplementary Note 1). Other clinical features of the patient included sparse hair and craniofacial disorders such as scaphocephaly, prominent forehead, and micrognathia. X-ray examination showed delayed bone age (two years old based on radiograph calculation) despite the generally normal spine and limbs (Supplementary Fig. 2a-d). Brain magnetic resonance imaging revealed hypomyelination of bilateral frontal lobes and clear septum cyst (Fig. 1a, b). While the pituitary gland appeared morphologically normal (Supplementary Fig. 2e, f), growth hormone deficiency was detected

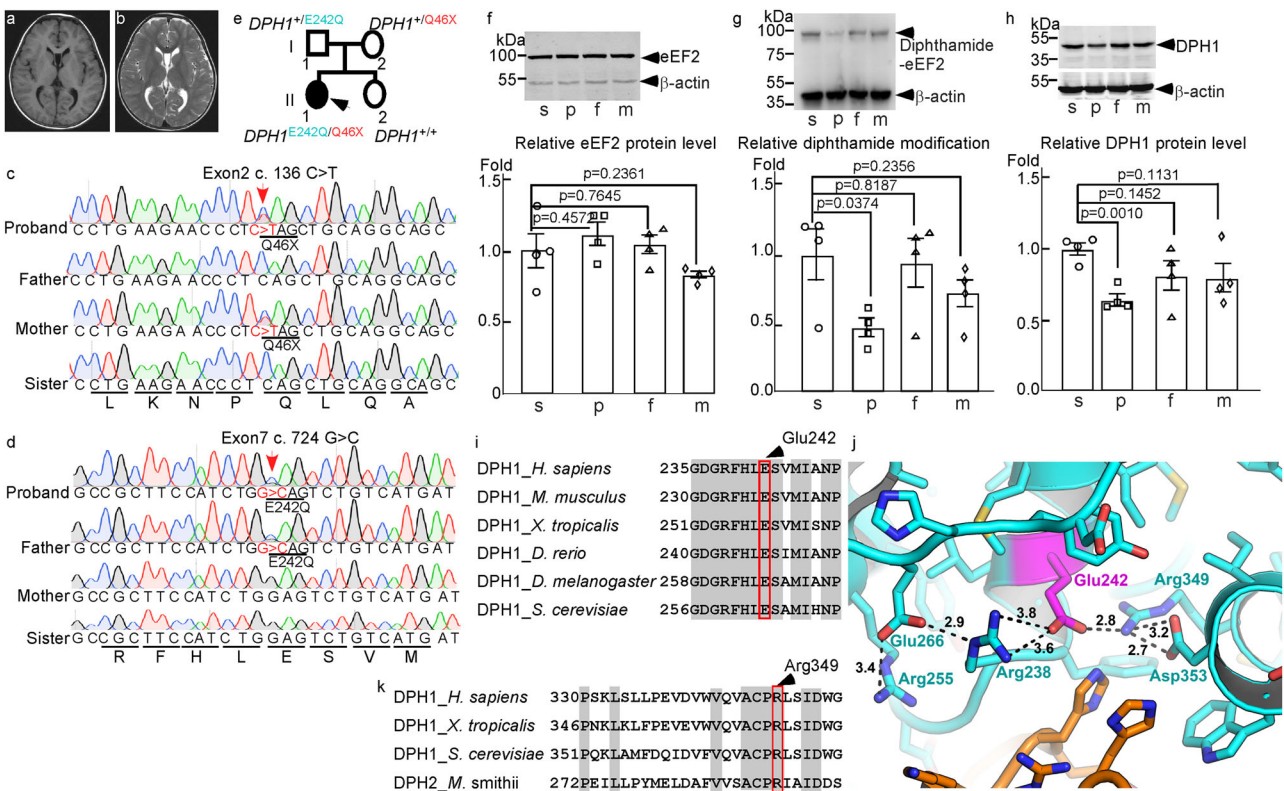

**Fig. 1 | Compound heterozygous *Dph1* mutations from a DEDSSH patient cause reduced eEF2 diphthamide modification.** T1(**a**) and T2 (**b**) weighted images of brain magnetic resonance imaging showing hypomyelination of bilateral frontal lobes and clear septum cyst. **c, d** Validation of *DPH1* mutations in the proband and her immediate family members by Sanger sequencing. Encoded amino acid sequences are shown at the bottom. **e** Pedigree diagram showing the mutations in *DPH1* within the proband's family (black arrow indicates the proband). Western blotting detected total eEF2 (**f**), diphthamide-modified eEF2 (**g**), and total DPH1 (**h**) in lymphoblastoid cell lysates from the sister (s), proband (p), father (f) and mother (m). Representative blots are shown at the top, and quantification of four independent experiments is summarized in the lower graphs. Values in (**f, g,** and **h**) represent means ± SEM, and statistical significance was determined by unpaired *t* tests with two-sided analysis. **i** Sequence alignment showing the conservation of Glu242 (highlighted) and the surrounding residues in eukaryotic DPH1 proteins. **j** AlphaFold model of the active site of human DPH1 (cyan) in complex with archaeal EF2 (brown) based on PDB 6Q2D. Note the positions of Glu242 and Arg349. Distances between residues are in Å. **k** Sequence alignment showing the conservation of Arg349 (highlighted) and the surrounding residues in eukaryotic DPH1 and archaeal DPH2.

(Supplementary Note 1), and growth hormone replacement therapy successfully restored growth (Supplementary Fig. 2g).

To understand the genetic basis of these defects, whole-exome sequencing was performed for the patient as well as her non-symptomatic parents and sister. The patient (proband) carried two mutated alleles of the *DPH1* gene: a c.136C>T (p.Q46X) nonsense allele, also present in the mother, and a c.724G>C (p.E242Q) missense allele, also present in the father; neither mutation was found in the sister (Fig. 1c–e). Although these mutations have not been reported previously, the proband was diagnosed with DEDSSH1 based on the similarities in symptoms with other patients with autosomal recessive *DPH1* mutations[4–8].

### Reduced DPH1 activity is associated with DEDSSH1 syndrome

DPH1 is required for catalyzing the diphthamide modification unique for eEF2[2]. To determine the effects of the proband's mutations on DPH1 function, lymphoblastoid cells were generated from the proband and other members of the family. While total eEF2 protein levels remained unchanged, diphthamide-modified eEF2 was reduced over 50% in the cells from the proband as compared with those from the sister that expressed wild-type DPH1; no significant reduction of eEF2 diphthamide modification was found in the cells derived from either parent (Fig. 1f, g). Surprisingly, a comparison of the endogenous DPH1 protein levels in mother- and sister-derived cells did not show any significant difference (Fig. 1h), despite the presence of a nonsense allele in the mother. This may be caused by enhanced expression of the wild-type allele in mother-derived cells to compensate for the loss of DPH1 expression due to the nonsense mutation. However, proband-derived cells contained significantly less DPH1 than sister-derived cells (~20% reduction; Fig. 1h), indicating that either the compensation or protein stability was compromised by the E242Q mutation.

Glu242 and adjacent residues are highly conserved in eukaryotic DPH1 (Fig. 1i). While there is no published structure for DPH1, structural modeling using AlphaFold v2.0[12] shows with high confidence that Glu242 forms a salt bridge with Arg349 (Fig. 1j), a residue conserved between eukaryotic DPH1 and archaeal DPH2 (Fig. 1k). The corresponding arginine residue forms a salt bridge with the SAM carboxyl group in the structure of archaeal DPH2, and substitution of this arginine with alanine completely abolishes ACP transfer activity without affecting SAM cleavage[3]. Hence the conserved Glu242 of eukaryotic DPH1 may play an important role in positioning Arg349 in the active site to interact with the SAM carboxyl group, and the E242Q mutation in the proband, which disrupts the Glu242-Arg349 salt bridge, leads to partial loss of DPH1 function. To examine if the other known DEDSSH-associated missense DPH1 mutations[5,6,8,13] also lead to decreased DPH1 activity, we generated *DPH1*-KO HEK293T cells using CRISPR-Cas9. As compared with wild-type DPH1, all tested mutants showed reduced abilities to rescue eEF2 diphthamide modification when overexpressed in *DPH1*-KO HEK293T cells (Supplementary Fig. 3a–h). Thus, we conclude that loss of DPH1 activity is associated with DEDSSH1 syndrome.

### DPH1 depletion decreases proliferation in the neuroepithelium and pre-migratory NC cells in *Xenopus* embryos

The NC cells are migratory and multipotent stem cells that have major contributions to hair follicles, craniofacial structures, and growth hormone-producing cells in the pituitary gland[14–16]. Therefore, some of the characteristics of DEDSSH, including sparse hair, craniofacial disorders and growth hormone deficiency, may be the consequences of general defects in NC development. In situ hybridization (ISH) showed that both *Dph1* and *eEf2* transcripts were broadly expressed in E9.5 mouse embryos, with high expression levels in the frontonasal mass, branchial arches and forelimb bud (Supplementary Fig. 4a, b), all of which contain newly arrived NC cells by this stage[17,18]. In *Xenopus* embryos, ISH detected high expression of *dph1* and *eef2* mRNA in the

early migrating NC at neurula stages and NC-populated branchial arches at tail bud stages (Supplementary Fig. 4c-i). Immunohistochemistry (IHC) also confirmed the enrichment of DPH1 protein expression in pre-migratory and early migrating NC (Supplementary Fig. 5a-f).

Due to the technical difficulties in studying early NC development in mice[19], we focused on *Xenopus*, an established model for this developmental process[20,21]. CRISPR/Cas9-based F0 mosaic KO is an efficient loss-of-function tool for *Xenopus*[22]. Co-injection of each of three guide RNAs targeting *dph1* (gRNAs g1-g3; Fig. 2a) with Cas9 protein into one blastomere of 2-cell stage *X. tropicalis* embryos reduced Dph1 expression levels on the injected side of nearly all embryos (Fig. 2b). Western blotting further confirmed the downregulation of DPH1 when embryos were injected at one-cell stage (Fig. 2c). Sanger sequencing of 10 randomly picked genomic DNA clones derived from embryos injected with g3 and Cas9 revealed insertions and deletions in essentially all clones leading to frameshift (Supplementary Fig. 6), confirming the high *dph1*-KO efficiency.

We have generated a transgenic *X. tropicalis* line expressing eGFP driven by the promoter of NC marker *snai2* for live imaging of NC development[23]. Injection of the gRNAs with Cas9 reduced eGFP expression in early migrating NC in the transgenic *snai2:eGFP* embryos (Fig. 2d), suggesting a decrease in NC population. Furthermore, injection of a translation-blocking morpholino (MO) for *dph1* decreased the expression of the NC marker *foxd3* and the neural plate border markers *pax3*, *msx1*, and *zic1* on the injected side prior to NC migration (stage -15; Fig. 2e). These data indicate that depletion of DPH1 decreased cells at the neural plate border, including pre-migratory NC cells. To examine cell proliferation, we carried out IHC for phosphorylated histone H3 (pH3), a widely used proliferation marker. At early neurula stage (stage -15), pH3 signals were primarily detected in the dorsal neuroepithelium (Fig. 2f), where cell proliferation is more active than in the ventral ectoderm[24]. Co-injection of g2 or g3 with Cas9 significantly reduced pH3 in the neuroepithelium on the injected side (Fig. 2f, g), indicating that DPH1 KO inhibited proliferation. Thus, a major function of DPH1 in the pre-migratory NC appears to be promoting cell proliferation. To determine if this function is dependent on eEF2, we tested the ability of various forms of eEF2 to rescue the reduced proliferation caused by loss of DPH1. While wild-type eEF2 failed to rescue, the G717R mutant that mimics diphthamide-modified eEF2[1,25] restored cell proliferation in most *dph1*-KO embryos (Fig. 2g). Taken together, our data suggest that DPH1-mediated eEF2 diphthamide modification is required for maintaining normal cell proliferation in the neuroepithelium.

### Knockin (KI) mice carrying the proband's mutations display DEDSSH phenotypes

To test directly if the proband's mutations in *DPH1* cause DEDSSH1 syndrome, we generated KI mice harboring either c.121C>T, p. Q41X, or c.709G>C, p. E237Q mutant allele, which corresponds to human Q46X and E242Q mutations, respectively (Supplementary Fig. 7a, b). The parental mice were then cross-bred to generate the *Dph1^E237Q/Q41X* double-mutant mice that mimic the proband's genetic background (Supplementary Fig. 7c). Both the *Dph1^Q41X/+* and *Dph1^E237Q/+* single KI mice were born at expected Mendelian ratio and appeared grossly normal, whereas all *Dph1^E237Q/Q41X* compound heterozygous KI mice died before birth with reduced body size that was apparent at E10.5 (Fig. 3a–c and Supplementary Table 1). As in *Xenopus* embryos with DPH1 depleted (Fig. 2f, g), IHC detected drastically decreased pH3 in *Dph1^E237Q/Q41X* embryos (Fig. 3d, e), suggesting that the reduced body size may be caused by proliferation defects. There was also a reduction in the NC marker SOX10 (Fig. 3f, g), which is in line with the reduced NC markers in *Xenopus* embryos (Fig. 2d, e) and likely as a consequence of the proliferation defects. At E14.5, the palatal shelves of *Dph1^E237Q/Q41X* embryos extended downwards as compared with wild-type embryos,

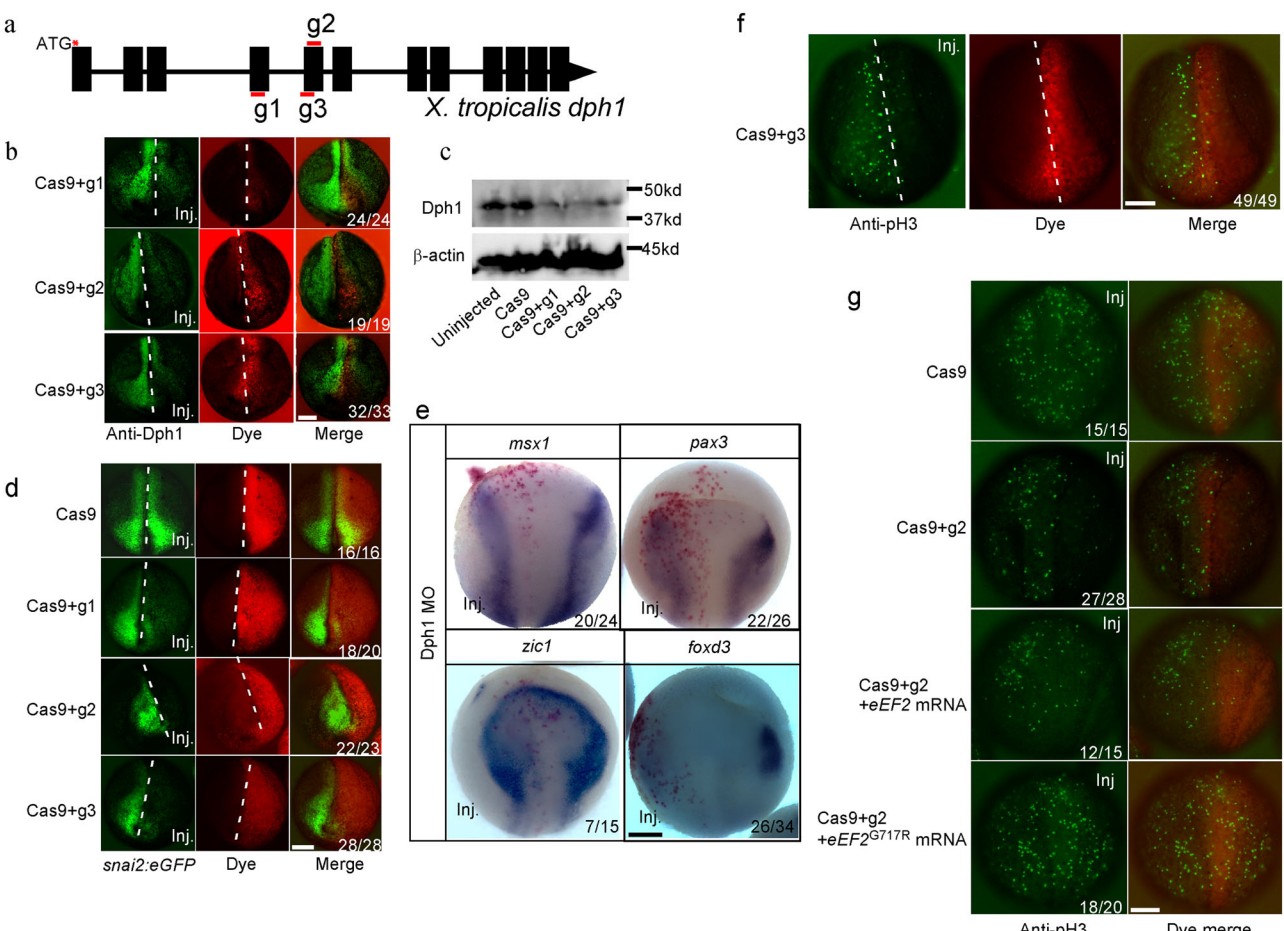

**Fig. 2 | Dph1 depletion in *X. tropicalis* embryos inhibits proliferation in the neuroepithelium and reduces pre-migratory NC by affecting eEF2 diphthamide modification. a** Design of gRNAs targeting *X. tropicalis dph1*. Black rectangles denote exons, red bars indicate gRNA target regions, and red asterisk highlights the translation start site. Wild-type (**b**) or *snai2-eGFP* transgenic (**d**) embryos were injected in one blastomere at 2-cell stage with the indicated Cas9 protein and gRNA, cultured to stage -17, and processed for IHC for DPH1 (**b**) or directly imaged for eGFP (**d**). Dextran 555 dye (red) was co-injected as a lineage tracer to identify the injected side. **c** Wild-type embryos were injected at one-cell stage with the indicated Cas9 protein and gRNA, and cultured to stage -17. Western blotting for DPH1 was carried out for whole-embryo lysates. **e** Wild-type embryos were injected in one blastomere at 2-cell stage with DPH1 MO, cultured to stage -12.5, and processed for ISH for the indicated mRNA. ß-galactosidase mRNA was co-injected as a lineage tracer, and red-gal staining (red) was performed to identify the injected side. **f, g** Wild-type embryos were injected in one blastomere at 2-cell stage with the indicated Cas9 protein, gRNA and *eef2* variants, cultured to stage -15, and processed for IHC for phosphorylated histone H3 (pH3). The injected side is denoted by co-injected Dextran 555 dye (red). The denominator and numerator represent the total number of embryos scored and the number of embryos with phenotypes similar to the image, respectively. All experiments were independently repeated three times. Scale bar, 100 μm.

as seen more clearly on the posterior side (Fig. 3h–k). While the palatal shelves completely fused in wild-type embryos at E15.5, they remained unfused in *Dph1^E237Q/Q41X^* embryos at this stage (Fig. 3l–o). Mandible of the *Dph1^E237Q/Q41X^* embryos also appeared to be smaller (Fig. 3p), and dissected Meckel's cartilage in the lower jaw was significantly shorter than the wild-type control (Fig. 3q–s). These growth defects and craniofacial phenotypes are consistent with the symptoms of the proband and are highly similar to the phenotypes of *Dph1*-KO mice[9,10], confirming that E237Q is a hypomorph mutation.

We next investigate how diphthamide deficiency affects cell proliferation. Phosphorylation of eEF2 by eEF2 kinase inhibits translation elongation[26]; however, KO of DPH1 in U251 glioblastoma cells using CRISPR/Cas9 did not change the levels of phosphorylated eEF2 (Supplementary Fig. 8). eEF2 is a key component of the ribosome, and disrupted ribosome biogenesis ("ribosomopathies") can stabilize the transcription factor p53, leading to NC hypoplasia as a consequence of reduced proliferation and increased apoptosis in the neural ectoderm[27–31]. In our DPH1-KO *Xenopus* embryos, cell proliferation was inhibited in the neuroepithelium (Fig. 3), but our TUNEL assays did not show any elevated apoptosis (Supplementary Fig. 9a). Similarly, we

could barely detect cleaved caspase-3 in both wild-type and *Dph1^E237Q/Q41X^* mouse embryos as well as craniofacial sections obtained from these embryos, and there was no apparent difference in the level of cleaved caspase-3 between the two groups of embryos (Supplementary Fig. 9b–e). Furthermore, p53 levels also remained unaltered (Supplementary Fig. 9d). Thus, the NC defects caused by loss of DPH1 cannot be attributed to increased p53 levels, as in the case of ribosomopathies[27–31].

As in *Dph1^E237Q/Q41X^* embryos and *Xenopus* neuroepithelium with DPH1 KO, KO of DPH1 reduced pH3 levels and inhibited proliferation in U251 cells (Fig. 4a and Supplementary Fig. 10a). There was a concomitant increase in cell size (Supplementary Fig. 10b–d), indicating cell cycle arrest. KO of DPH1 increased cell cycle inhibitor p21, despite a reduction of p53 levels (Fig. 4a). Since p53 is well known to induce *p21* transcription, the reduced p53 could be due to negative feedback after prolonged p21 induction in the DPH1-KO cells. To clarify this issue, we carried out transient knockdown (KD) of DPH1 using two separate siRNAs. KD of DPH1 increased the protein and mRNA levels of p21 and PUMA, another p53 target, without affecting p53 (Fig. 4b, c). However, mRNA levels of the other p53 targets, including *ALDH4*, *BAX,* and

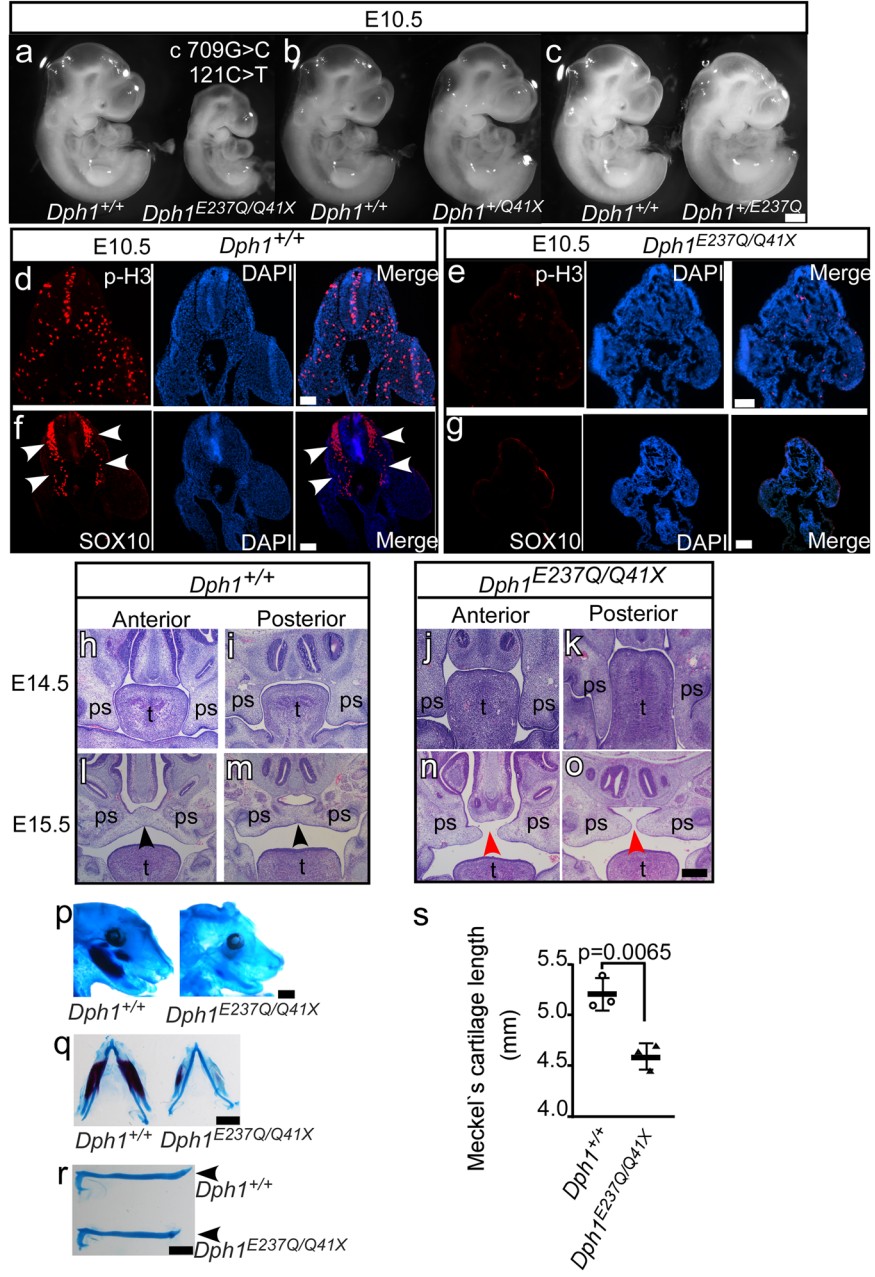

**Fig. 3 | KI mouse embryos carrying the proband's *Dph1* mutations have growth retardation and craniofacial defects. a–c** The *Dph1^{E237Q/Q41X}* compound heterozygous embryos have reduced body size, as compared with wild-type littermates and littermates harboring only one mutated allele, at E10.5 (numbers of biologically independent mice examined over 21 experiments: *Dph1^{E237Q/Q41X}*, n = 28; *Dph1^{E237Q/+}*, n = 46; *Dph1^{Q41X/+}* n = 44; *Dph1^{+/+}*, n = 45). Scale bar, 250 μm. IHC for pH3 (**d, e**) and SOX10 (**f, g**) in wild-type (**d, f**) and *Dph1^{E237Q/Q41X}* (**e, g**) embryos at E10.5 (numbers of biologically independent samples: *Dph1^{E237Q/Q41X}*, n = 3; *Dph1^{+/+}*, n = 3). Scale bar, 100 μm. **h–o** Palatal defects in *Dph1^{E237Q/Q41X}* embryos, as shown by H&E-stained sections of anterior and posterior portions of palatal shelves from embryo heads (numbers of biologically independent samples: *Dph1^{E237Q/Q41X}*, n = 4, *Dph1^{+/+}*, n = 3).

The *Dph1^{E237Q/Q41X}* embryos have downward extension of the palatal shelves (ps) relative to the tongue (t) at E14.5 (**j, k**), and unfused palatal shelves at E15.5 (red arrowheads in **n, o**). Scale bar, 100 μm. Mandible hypoplasia in *Dph1^{E237Q/Q41X}* embryos, as indicated by Alcian blue (cartilage) and Alizarin red (bone) double staining. **p** Side view of the craniofacial cartilage and bones. Scale bar, 1000 μm. Dissected Meckel's cartilage shown on both sides (**q**) and single side (**r**), and quantified in (**s**) (n = 3 biologically independent samples). Values in (**s**) represent means ± SEM of three biologically independent samples, and statistical significance was determined by unpaired *t* test with two-sided analysis. Scale bar, 1000 μm (**q**) and 800 μm (**r**).

---

*MDM2*, were not significantly changed (Fig. 4c). In E10.5 *Dph1^{E237Q/Q41X}* mouse embryos, we also observed increased p21 protein levels along with decreased pH3 (Fig. 4d), and RT-qPCR detected significantly elevated mRNA levels of *p21* and *PUMA* but not the other p53 targets examined (Fig. 4e). The lack of influence on these other p53 targets confirms that loss of DPH1 does not stabilize p53 as in ribosomopathies; instead, it likely boosts p53 transcriptional activity for specific target genes such as *p21* and *PUMA*. Similarly, KD of DPH2 and 6, two

other essential components of the diphthamide biosynthesis pathway[1,2], also led to significantly elevated p21 in U251 cells (Supplementary Fig. 11a, b), indicating that this effect is a general consequence of diphthamide deficiency.

Since eEF2 is the only protein known to undergo diphthamide modification, we asked if eEF2 can interact with p53. Indeed, co-immunoprecipitation (co-IP) detected an association between endogenous p53 and eEF2 in U251 cells, which was enhanced by DPH1 KD

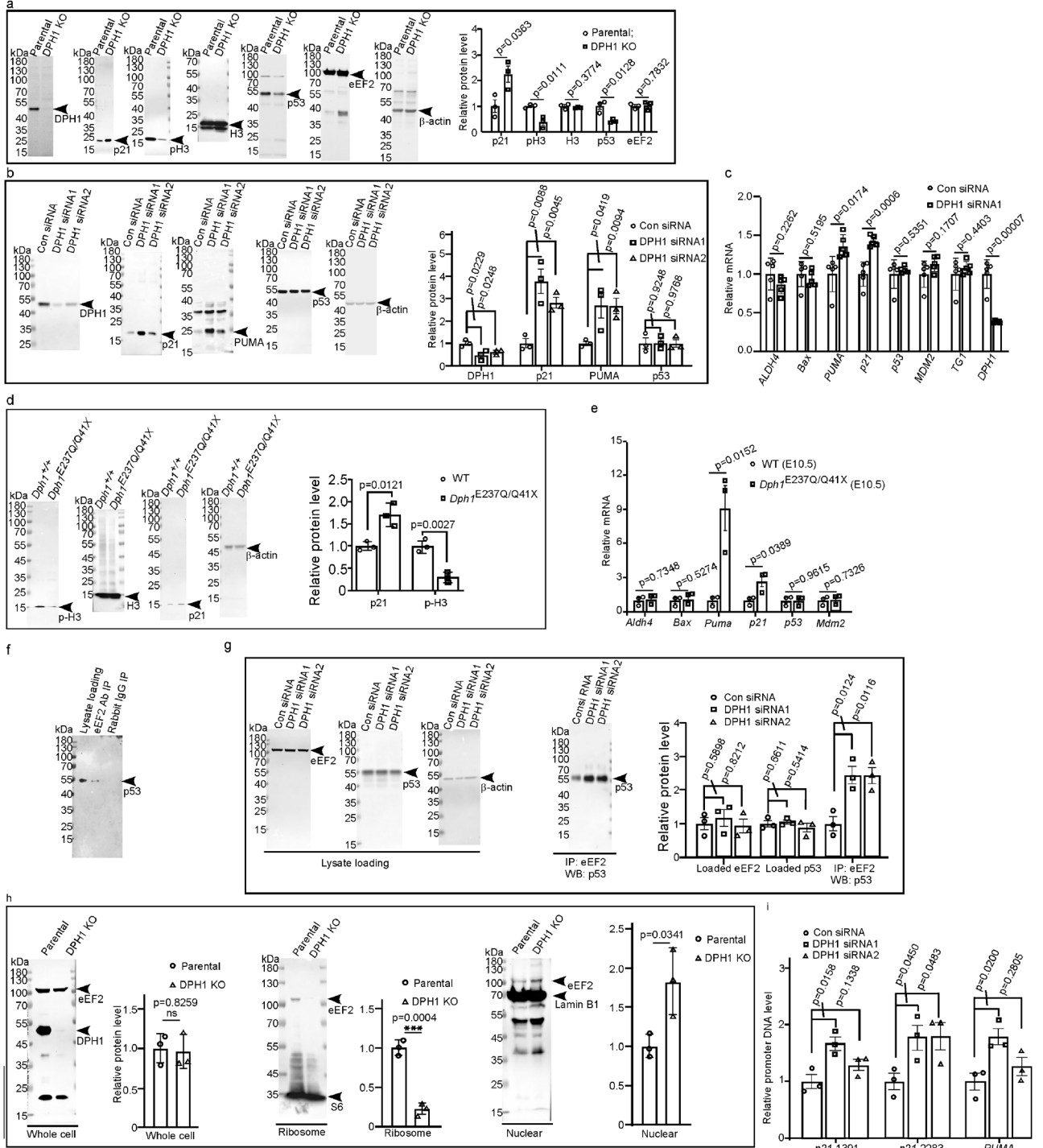

**Fig. 4 | Loss of DPH1 function upregulates p21 and PUMA by promoting eEF2-p53 complex formation and p53 binding to *p21* and *PUMA* promoters.** Western blotting for the indicated proteins was carried out for the lysates of parental or *DPH1*-KO U251 cells (**a**), or U251 cells transfected with the indicated siRNA (**b**). Representative blots are shown on the left, and the quantification of *n* = 3 independent experiments is summarized in the graphs on the right. **c** RT-qPCR results for the indicated genes in U251 cells with and without DPH1 KD, *n* = 5 independent experiments. **d**, **e** Western blotting (*n* = 3 biologically independent mice) and RT-qPCR results (*n* = 3 biologically independent mice) for the indicated proteins and genes, respectively, in E10.5 wild-type (WT) and *Dph1*^E237Q/Q41X mouse embryos. Co-IP of eEF2 and p53 in U251 cells. Lysates of untreated U251 cells (**f**) or cells transfected

with the indicated siRNA (**g**) were processed for IP with an anti-eEF2 antibody, and western blotting (WB) was carried out for p53 and eEF2 (as control; *n* = 3 independent experiments). **h** Comparison of the amount of eEF2 in whole-cell lysates as well as ribosomal and nuclear fractions of parental and DPH1 KO U251 cells, as indicated, by western blotting (*n* = 3 independent experiments). **i** U251 cells were transfected with the indicated siRNA, and ChIP-qPCR was performed using a p53 antibody to quantify the amount of indicated promoter fragment associated with p53 (*n* = 3 independent experiments). Values represent means ± SEM in (**a, b, c, d, e, g, h, i**), and statistical significance was determined by unpaired *t* test with two-sided analysis.

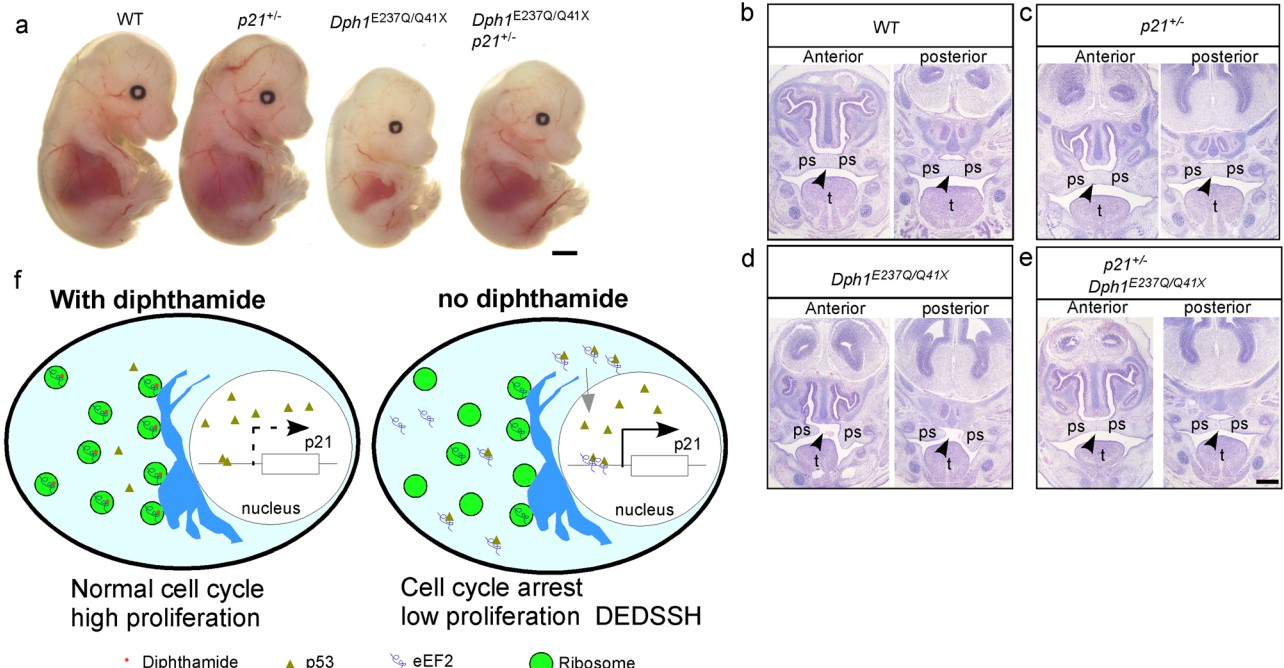

**Fig. 5 | Loss of one *p21* allele rescues the growth and craniofacial phenotypes in *Dph1*[E237Q/Q41X] embryos. a** Comparison of a *Dph1*[E237Q/Q41X] *p21*[+/−] embryo with wild-type (WT), *p21*[+/−], and *Dph1*[E237Q/Q41X] littermates at E15.5. While all *Dph1*[E237Q/Q41X] embryos show greatly reduced body size (*n* = 4), 67% of *Dph1*[E237Q/Q41X] *p21*[+/−] embryos (*n* = 9) exhibited enlarged body size as compare with their *Dph1*[E237Q/Q41X] littermates.

Scale bar, 1000 μm. **b-e** At E15.5, palate was fused in 100% of wild-type (*n* = 4) or p21[+/−] (*n* = 5) embryos, 0% of *Dph1*[E237Q/Q41X] (*n* = 4) embryos, and 80% of *Dph1*[E237Q/Q41X] *p21*[+/−] embryos (*n* = 5). Scale bar, 250 μm. **f** A schematic diagram illustrates how diphthamide modification affects cell proliferation.

(Fig. 4f, g). eEF2 has six domains, the three N-terminal domains containing GTPase activity, and the three C-terminal domains that mimic tRNA structure[32,33]. Exogenously expressed HA-tagged full-length eEF2 and a truncated form containing the three N-terminal domains (eEF2[1-486]), but not a truncated form containing the three C-terminal domains (eEF2[487-858]), were associated with endogenous p53 (Supplementary Fig. 12a, b), suggesting that the N-terminal GTPase domains are required and sufficient for p53 binding. eEF2 is primarily localized in the cytoplasm especially in the ribosome, but can also enter the nucleus[34,35]. Upon DPH1 KO in U251 cells, there was significantly less eEF2 protein associated with the ribosome but more in the nucleus (Fig. 4h). Similarly, in patient-derived *Dph1*[E242Q/Q46X] lymphoblastoid cells, there was more nuclear eEF2 as compared with the *Dph1*[+/+] cells from the sister (Supplementary Fig. 13a, b). Thus, diphthamide deficiency reduces the association of eEF2 with the ribosome while promoting its nuclear translocation and interaction with p53.

To examine whether the increased eEF2-p53 interaction leads to recruitment of more p53 to *p21* and *PUMA* promoters, chromatin-IP (ChIP)-qPCR was performed to quantify the binding of p53 to these promoters. There are two p53 binding sites in the *p21* promoter, a high-affinity site at −−2283 and a low-affinity site at −−1391. The *PUMA* promoter also contains two p53 binding sites, but both are localized within a single 92-bp region[36]. KD of DPH1 enhanced the binding of p53 to the *p21* and PUMA promoters (Fig. 4i). These data suggest that loss of diphthamide modification facilitates eEF2/p53 complex formation and binding of p53 to the *p21* promoter, generating more p21 to inhibit cell proliferation.

**Deletion of one *p21* allele rescues the DEDSSH phenotypes of the KI mice**

Finally, we tested if reduction of p21 can rescue the DEDSSH phenotypes caused by loss of DPH1 function. To this end, we generated heterozygous *p21* KO mice, which contained about half the amount of p21 protein as compared with wild-type littermates (Supplementary

Fig. 14a–d), and cross-bred them with the *Dph1*[E237Q/Q41X] mutants (Supplementary Table 2). While loss of one *p21* allele did not cause any apparent phenotypes in the wild-type background, it partially rescued the reduced body size in *Dph1*[E237Q/Q41X] mutants (Fig. 5a; *P* < 0.001). In contrast to *Dph1*[E237Q/Q41X] embryos, which had cleft palates at E15.5, the palatal shelves fused in most *Dph1*[E237Q/Q41X] *p21*[+/−] embryos at this stage (Fig. 5b-e; *P* < 0.001), indicating that elevated p21 expression contributes to these DEDSSH phenotypes.

## Discussion

We have identified novel compound heterozygous mutations in *Dph1* causing DEDSSH by impairing eEF2 diphthamide modification, which leads to redistribution of eEF2 from ribosomes to p53. This redistribution could be due to enhanced binding affinity of unmodified eEF2 for p53, or reduced affinity for ribosomes (as diphthamide is localized at the tip of eEF2 domain IV and interacts with the mRNA and the ribosome[37–39]), or both. Consequently, there is more eEF2-bound p53 with enhanced association with the *p21* promoter to induce more p21 expression, causing inhibited NC proliferation (Fig. 5f). These results reveal a critical role of eEF2 diphthamide modification in NC development and the mechanisms underlying DEDSSH.

Because eEF2 is essential for ribosome translocation, DEDSSH has been proposed to be a ribosomopathy[4]. A hallmark of ribosomopathies is p53 stabilization[27–31], which is thought to be caused by the inhibition of MDM2, the E3 ubiquitin ligase for p53, by free ribosomal proteins[40,41]. Subsequent activation of the p53 pathway leads to decreased proliferation and increased apoptosis, especially in the neuroepithelium[27–31]. Although a previous study showed that loss of p53 partially rescued the growth and cleft palate phenotypes in *DPH1*-KO mice, no general p53 activation was found in *DPH1*[−/−] MEF cells[9]. We also did not detect p53 stabilization or enhanced transcription of most p53 target genes in *DPH1*-KO/KD cells or *Dph1*[E237Q/Q41X] embryos (Fig. 4a–e). Furthermore, diphthamide-deficient eEF2 can still function in ribosome translocation[2], suggesting that ribosome structure remains

intact. Our data instead point to an enhanced association between eEF2 and p53 in the absence of diphthamide modification, which promotes the recruitment of p53 to distinct targets such as the *p21* promoter. The mechanism underlying this recruitment remains unclear, but p53-mediated transcription of distinct downstream targets such as *p21* has been shown to be selectively regulated. For example, mutations of certain amino acid residues in p53 involved in transactivation or acetylation differentially affect target gene transcription[42,43], and binding of p53 to the *p21* promoter can be regulated by epigenetic events and non-coding RNAs[44,45]. P21 KO largely rescues the growth and craniofacial deficiencies in $Dph1^{E237Q/Q41X}$ mice (Fig. 5a–e), providing further evidence supporting that the enhanced p21 expression contributes to DEDSSH etiology. Another p53 target upregulated upon loss of DPH1 function is PUMA (Fig. 4b–d), which may induce apoptosis. However, unlike ribosomopathies, no significant induction of apoptosis was observed in cells or mouse embryos with loss of DPH1 function. In addition to *PUMA*, p53 can induce the expression of a number of pro-apoptotic genes such as *NOXA*[46], and cytoplasmic p53 also has PUMA-independent function in initiating apoptosis[47]. It is therefore not surprising that less apoptosis is observed in diphthamide deficiency than in ribosomopathies, which have upregulated p53 levels. Overall, our data suggest that there are distinct mechanistic differences between diphthamide deficiency and ribosomopathies.

Both *S. cerevisiae* and MCF7 cells are resistant to diphthamide deficiency-induced growth arrest[2,11]. Interestingly, *S. cerevisiae* does not have a p53 homolog[48], and MCF7 cells contain high levels of cyclin Ds that can overcome the inhibition of cell cycle progression by p21[49]. In contrast, NC cells express substantially higher levels of *p53* transcripts and are particularly susceptible to p53-mediated responses[27]. Thus, the cell type-specific proliferation effects caused by diphthamide deficiency may reflect different cells' sensitivity to the p53-p21 pathway.

Growth retardation and short stature are common symptoms of NC defects[21,50], but the relationship between NC and growth is unclear. In the patient with compound heterozygous *Dph1* mutations and NC defects, we uncovered growth hormone deficiency, which was treatable with growth hormone replacement therapy (see Patient Description and Fig. S2g). Because the hormone-producing cells in the adenohypophysis of the pituitary gland derive from the NC[16], these results provide a connection between NC defects and growth retardation. In mice, the adenohypophysis starts to secret growth hormone around E15.5[51]. Thus, our data suggest that the smaller size of diphthamide-deficient embryos is mainly caused by reduced proliferation, whereas growth hormone deficiency has a major contribution to the growth retardation in humans after birth. It would be of interest to test if similar growth hormone replacement therapies can apply to other patients with growth retardation caused by NC defects. In conclusion, our study offers not only mechanistic insights into the roles of diphthamide deficiency in NC defects, but also therapeutic options for affected patients.

## Methods

### Plasmids and reagents

The cDNAs for *Xenopus tropicalis dph1*, *eef2*, mouse *Dph1* and *eEf2* were amplified from corresponding embryos and subcloned into pCS2+ expression vector. The cDNA for human eEF2 was amplified from U251 cells and subcloned into pCS2+ vector to generate full-length eEF2 as well as the truncated forms $eEF2^{1-486}$ and $eEF2^{487-858}$ with C-terminal HA tag. All of the PCR primers used in this work were listed in Supplementary Table 3. Templates for in situ hybridization for *snai2*, *sox10*, *pax3*, *zic1*, *msx1*, *foxd3* were amplified from stage -14 *Xenopus tropicalis* embryo cDNA library using primers listed in Supplementary Table 3, with T7 promoter sequence incorporated by PCR. The in situ hybridization probes were then transcribed in vitro using a MAXIscript™ T7 transcription kit (Invitrogen). Antibodies used for

western blotting (WB), immunohistochemistry (IHC) and immunoprecipitation (IP) include rabbit anti-OVCA1/DPH1 (Abcam ab185960, 1:3000 for WB), rabbit anti-DPH1 for *Xenopus* embryos (Lifespan Bioscience LS-C116434, 1:3000 for WB, 1:200 for IHC), rabbit anti-eEF2 (Cell Signaling Technology 2332, 1:3000 for WB, 1:200 for IHC), rabbit anti-eEF2 (Proteintech 20107-1-AP, 1:60 for IP), rabbit anti-phospho-eEF2 (Thr56)(Cell Signaling Technology 2331, 1:3000 for WB), mouse anti-HA (Abclonal AE008, 1:50 for IP), rabbit anti-HA (Cell Signaling Technology 3724, 1:3000 for WB), rabbit anti-Sox10 (Cell Signaling Technology 89356, 1:200 for IHC), HRP-conjugated anti-beta-actin (Biodragon BD-PB3508, 1:10,000 for WB), rabbit anti-phospho histone3 (Cell Signaling Technology 53348, 1:3,000 for WB, 1:200 for IHC), rabbit anti-histone H3 (Ser10)(Proteintech 17168-1-AP, 1:5000 for WB), rabbit anti-p53 (Proteintech 10442-1-AP, 1:3,000 for WB), mouse anti-p53 (Cell Signaling Technology 488185, 1:2000 for WB), rabbit anti-p21(Proteintech 103551-1-AP, 1:3000 for WB with U251 cell lysate), rabbit anti-p21(Abclonal A19094, 1:3000 for WB with mouse embryo), rabbit anti-PUMA (Proteintech 55120-1-AP, 1:3000 for WB), rabbit anti-caspase3 (Proteintech 19677-1-AP, 1:3,000 for WB, 1:200 for IHC), rabbit anti-cleaved caspase3 (Asp175)(Cell Signaling Technology 9661, 1:3000 for WB, 1:200 for IHC), rabbit anti-lamin B1 (Proteintech 12987-1-AP, 1:5000 for WB), rabbit anti-S6 ribosome protein (Cell Signaling Technology 2217, 1:3000 for WB), HRP-conjugated goat anti-rabbit IgG (Beyotime A0208, 1:10,000 for WB), and Cy3-conjugated goat anti-rabbit IgG (Beyotime A0516, 1:200 for IHC). siRNA duplexes were designed and synthesized by GenePharma, and the sequences are shown in Supplementary Table 4.

### Generation of *Dph1* KI mice

KI mice with *Dph1* c.709G>C (E237Q) and c.121C>T (Q41X) single mutations were designed and generated by Shanghai Model Organisms Center, Inc. (Shanghai, China). Cas9 mRNA was transcribed in vitro with mMESSAGE mMACHINE T7 Ultra Kit (Ambion, TX, USA) according to the manufacturer's instructions, and subsequently purified using the MEGAclear™ kit (Invitrogen). sgRNAs (5'-TGGAGATGGCCGCTTTCATC-3' for E237Q and 5'-CCAGTTACAGGCTGCTGTCCAAG-3' for Q41X, respectively) was transcribed in vitro using the MEGAshortscript Kit (Invitrogen) and subsequently purified using the MEGAclear™ kit. The transcribed Cas9 mRNA and sgRNA as well as a 110-base pair single-stranded oligodeoxynucleotide (ssODN) were co-injected into zygotes of C57BL/6J mice. Sequences of the ssODNs were:

5'CACTTACACACTCTTCCCACCCTGCAGGTATCTTGGAGATGGC CGCTTTCATCTGCAGTCTGTCATGATTGCCAACCCTAATATACCTGCT TACCGGTATGGGCTGGGGA3' (for generating the c.709G>C (E237Q) mutation), and

5'CATCTCTCGGGGGCGACTGGCCAATCAGATCCCCCCTGAGGTC CTGAACAACCCCTAGTTACAGGCTGCTGTCCAAGTTCTGCCTTCTAA CTACAACTTTGAGATCCCCA −3' (for generating the c.121C>T (Q41X) mutation).

Obtained F0 mice were genotyped by PCR and Sanger sequencing (see Supplementary Table 3 for primer sequences). The F0 mice with expected single mutations were chosen and crossed with wild-type C57BL/6J mice to produce the F1 progeny. The $Dph1^{Q41X/E237Q}$ compound heterozygous mutants were generated by crossing the c.709G>C (E237Q) with c.121C>T (Q41X) single KI mice, and genotyped.

### Generation of p21 KO mice

p21 KO mice was generated by Cyagen Biosciences Inc. (Suzhou, China) using CRISPR-Cas9 by delete Exon 2 with guide RNAs targeting GAGTTAATCACCAAGACAGC<u>AGG</u>   and   TGGCAGTAGAGCTCTAA-GAA<u>GGG</u>. (PAM sites are underscored). Genotyping was carried out by PCR using primers flanking Exon 2, generating 1476-bp and 501-bp products for wild-type *p21* and the deletion mutant, respectively (Supplementary Fig. 14).

## X. tropicalis embryo manipulation

Wild-type *X. tropicalis* frogs were purchased from NASCO, and the *snai2:eGFP* transgenic line was generated previously[23]. mRNAs encoding desired proteins were generated by in vitro transcription. For embryo production, female and male frogs were each primed with 20 IU human chorionic gonadotropin (HCG), and boosted with 200 IU HCG the next day for natural mating. Fertilized embryos were collected and injected in one blastomere at 2-cell stage; the uninjected blastomere served as a control. Alexa Fluor 555 (Invitrogen) or ß-galactosidase mRNA was co-injected as a lineage tracer. Morpholino for DPH1 KD (5′-CTTCCGCCATCTCTGACATATTTA-3′) was designed and synthesized by Gene Tools. The guide RNAs for CRISPR/Cas9 mediated DPH1 KO in *X. tropicalis* were designed using the CRISPRscan software[52]. Three individual sites were designed for CRISPR targeting. Protospacers g1 (5′-GTGATGGGCGATGTGACGTA-3′), g2 (5′-GGTTGAAAGTTGAAGCGAA-3′) and g3 (5′-GGCATCAATCGGGACT-GAG-3′) were embedded in forward primers. The DNA templates for gRNA transcription were obtained by PCR using pUC57-T7-gRNA scaffold vector as a template (see Supplementary Table 3 for primer sequences), and gRNAs were generated by in vitro transcription using the TranscriptAid T7 kit (Thermo Scientific K0441). To knock out *dph1*, each *X. tropicalis* embryo was injected with 1 ng Cas9 protein (PNA Bio) and 300 pg gRNA.

## Mouse craniofacial cartilage and bone characterization

Mouse embryos were harvested at E15.5, and stained with 0.1% Alcian blue and 1% Alizarin red[53]. Meckel's cartilage was dissected, photographed with a NIKON SMZ25 stereomicroscope, and measured with the NIS-Elements software.

## ISH and immunostaining

Mouse or *X. tropicalis* embryos were harvested and fixed with paraformaldehyde at desired stages and processed for in situ hybridization (ISH) or immunohistochemistry (IHC). For sections, embryos were embedded in wax and sectioned at 10 μm thick, followed by H&E or antibody staining. Cells were fixed with 4% paraformaldehyde for 20 min and permeabilized with 0.2% Triton-X100, followed by antibodies staining. Embryo images were taken with a Zeiss Axiozoom.v16 or Nikon SMZ25 epifluorescence microscope, and *Xenopus* embryos were scored by comparing the injected side with the uninjected side of the same embryos. Cell and section images were taken with an Olympus VS200 microscope.

## Cell line generation and maintenance

Lymphoblastoid cell lines were generated using Epstein-Barr virus (EBV) from the culture filtrate of EBV-transformed B95-8 marmoset cells[54]. Briefly, 3 ml whole blood was mixed with an equal volume of RPMI1640, added carefully on top of 6 ml lymphocyte separation medium (TBD Science LTS1077), and centrifuged at $800 \times g$ for 20 min. The intermediate white layer containing lymphocytes was carefully removed and washed twice by mixing with 10 ml RPMI1640 and centrifuging at $200 \times g$ for 10 min. The lymphocytes were incubated with 3 ml culture filtrate of EBV-transformed B95-8 cells for 15 min, subsequently mixed with complete culture medium (RPMI1640 supplemented with 10% fetal bovine serum (FBS)) and cultured at 37 °C with 5% $CO_2$ for one week. Transformed lymphoblastoid cells were round or oval, appeared larger than normal lymphocytes, and formed cell aggregates. These cells underwent robust proliferation after culture medium was replaced with fresh 2−3 times.

U251 and HEK293T cells (ATCC) were cultured in EMEM or DMEM, respectively, supplemented with 10% FBS at 37 °C with 5% $CO_2$. To knock out DPH1 in human cell lines, protospacer 5′-CAGGCAGGT-GATGGCGGCGC-3′ was subcloned into pX458 constructs with Bbsi[55] and transfected into cells for KO screening. Because diphthamide-deficient cells are resistant to Pseudomonas exotoxin A (PE) toxin,

100 nM PE was utilized to select for DPH1 KO cells. To knock down DPH genes, Lipofectamine RNAiMAX (Invitrogen) was applied with specific double-strand siRNA according to the manufacture guideline. The siRNA sequences are listed in Supplementary Table 4.

For cell proliferation assays, parental and DPH1$^{-/-}$ U251 cells were seeded into six-well dish at $10^5$/well. Cells were harvested and counted with cell counting slides (Bio-Rad 1450015) using a Bio-Rad TC20 automated cell counter.

For cell size comparison, parental and *DPH1$^{-/-}$* U251 cells were trypsinized and resuspended in PBS. 10 μl cell suspension was pipetted into the cell counting slides (Bio-Rad 1450015), and diameter of single cells was measured with Nikon NIS-Elements AR Analysis software.

To evaluate eEF2 nuclear localization, 10 μl lymphoblastoid cell suspension from the patient or her sister was smeared on a glass microscope slide. The single-layer cell samples were fixed and immunofluorescent staining was carried out with rabbit anti-eEF2 antibody (Cell Signaling Technology 2332) and Cy3-conjugated goat-anti-rabbit secondary antibody (Beyotime A0516). Nuclei were counterstained by DAPI. Nuclear eEF2 intensity was calculated by measuring the fluorescent intensity of eEF2 signal in DAPI-positive area using Nikon NIS-Elements AR Analysis software.

## Immunoprecipitation (IP), cell fractionation, and western blotting

Cells or mouse embryos were lysed in ice-cold RIPA buffer (Thermo Scientific 89901) plus Halt protease and phosphatase inhibitor cocktail (Thermo Scientific 78440), and processed for western blotting. Blots were detected with HRP-conjugated antibodies and chemiluminescence substrates using a Bio-Rad ChemiDoc imager. Band intensity was quantified with Quantity One, statistics was performed with unpaired *t* tests with two-sided analysis, and p values were calculated using the SPSS software.

For IP, Pierce Crosslink Magnetic IP/Co-IP kit (Thermo Scientific 88805) was used. Briefly, IP antibody was crosslinked to the magnetic beads, and the associated proteins were eluted according to manufacturer's instruction. For cell fractionation, cells were lysed in lysis buffer (10 mM KCl, 5 mM $MgCl_2$, 50 mM Tris-Cl, pH 7.4, 0.7% NP40, and protease inhibitor (Beyotime P1005)) on ice for 30 min, followed by centrifugation for 10 min at $750\,g$, 4 °C to pellet nuclei. The supernatant was then centrifuged for 10 min at $12,500 \times g$, 4 °C to pellet mitochondria. Ribosome fraction was obtained by ultra-centrifugation for 2 hours at $250,000 \times g$, 4 °C in sucrose cushion (1 M sucrose, 10 mM KCl, 5 mM $MgCl_2$, and 50 mM Tris-Cl, pH7.4)[56].

To quantify diphthamide modification, 10 μl cell lysate was incubated with 30 μl of mixture containing 100 nM PE toxin and 5 uM biotinylated NAD for 1 h at 25 °C. Only eEF2 with diphthamide modification can be biotinylated via the transfer of biotinylated ADP. Western blotting was subsequently performed using HRP-conjugated streptavidin[57].

## RT-qPCR and chromatin IP (ChIP)-qPCR

ChIP was carried out using the SimpleChIP Enzymatic Chromatin IP kit (Cell Signaling Technology 9003) according to manufactures instruction. For reverse transcription, total RNA was isolated from cultured cells or mouse embryos using the RNeasy Mini Kit (Qiagen), and reverse transcribed into cDNA using the RevertAid RT Kit (Thermo Scientific K1691). Quantitative PCR was performed on the Quant Studio 6 Flex (Applied Biosystems) using the Rotor-Gene SYBR Green Kit (Qiagen) and primers listed in Supplementary Table 3.

## Whole-exon sequencing

Whole-exon sequencing was performed by MyGenostics, Inc., Beijing, China. Briefly, genomic DNA was extracted from the patient's peripheral blood using a DNA extraction kit (DNeasy Blood & Tissue Kit, Qiagen) according to the manufacturer's instructions. The indexed

library was constructed using 3 μg DNA following manufacturer's protocol (MyGenostics, Inc., Beijing, China). The enriched library was sequenced on an Illumina HiSeq 2500 sequencer (Illumina, San Diego, CA, USA). The physical coverage and Q30 score were 99.62% and 0.891, respectively.

Base calling and quality assessment of sequencing reads were processed using Bcl2Fastq 2.18.0.12 (Illumina). Alignment of clean sequencing reads to the reference human genome (UCSC Genome Browser hg19) was run on the Short Oligonucleotide Analysis Package (SOAP) aligner software (SOAP2.21; soap.genomics.org.cn/soapsnp.html). All the single nucleotide variants and insertion-deletions (InDels) were identified by the Exome assistant program (http://122.228.158.106/exomeassistant). The *DPH1* variants were validated by sanger sequence of blood DNA PCR products, with the primers shown in Supplementary Table 3.

### Ethics statement

The usage of clinical samples for scientific research and clinical diagnosis was approved and authorized by the patients and ethics committee of Children's Hospital of Chongqing Medical University. Consent from the patient's guardian has been obtained for both participation in the study and the publication of patient information, including MRI images, genetic mutation information, and clinical information as described in Supplementary Note 1. Methods involving live animals were carried out in accordance with the guidelines and regulations approved and enforced by the Institutional Animal Care and Use Committees at the University of Delaware and Children's Hospital of Chongqing Medical University.

### Reporting summary

Further information on research design is available in the Nature Portfolio Reporting Summary linked to this article.

## Data availability

The source data for Figs. 1–4 and supplementary Fig. 3,8–11,13,14 are provided as a Source Data file. Whole exome sequencing data were deposited in a public WES database under accession HRA007036 and available by approved request due to patient privacy concerns. All other data that support the findings of this study are available upon reasonable request by emailing Yu Shi at shiyu@hospital.cqmu.edu.cn. Source data are provided with this paper.

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

## Acknowledgements

This work was supported by grants from Senior Medical Talents Program of Chongqing for Yong and Middle-aged to Y.S., Future Medical Youth Innovation Team Development Support Project of Chongqing Medical University (W0110 to C.S. and Y.S., Co-PI), Chongqing Science and Technology Committee (CSTB2023NSCQ-MSX0181 to D.H.), US NIH R01DE029802 to S.W., Science and Technology Research Project of Chongqing Education Commission (KJQN201900448 to C.S.), General program of Clinical Medical Center, National Clinical Research Center for Child Health and Disorders (NCRCCHD-2020-GP-10 to C.S.), the fundings from the National Natural Science Foundation of China (82230043 and 82293642), the Key Laboratory of Alzheimer's Disease Of Zhejiang Province, and Oujiang Laboratory (W.S.).

## Author contributions

Yu Shi, Shuo Wei and Weihong Song designed the experiments. Yu Shi, Daochao Huang, Cui Song, Ruixue Cao, Zhao Wang, Dan Wang, Li Zhao, Xiaolu Xu, Congyu Lu, Feng Xiong, Haowen Zhao, Shuxiang Li, Quansheng Zhou, Shuyue Luo, Dongjie Hu, Yun Zhang, Cui Wang, Yiping Shen, Weiting Su, Yili Wu and Karl Schmitz performed the experiments and analyzed the data. Yu Shi, Shuo Wei and Weihong Song wrote the manuscript. Shuo Wei and Weihong Song conceived and supervised the project.

## Competing interests

The authors declare no competing interests.
