## [Peer Review File · Nature Communications]

Diphthamide deficiency promotes association of eEF2 with p53 to induce p21 expression and neural crest defectsReviewer #1 (Remarks to the Author):

Diphthamide is a modified Histidine residue on eEF2, with its modification depends on a protein complex including Dph1 to Dph7, as well as a regulator Miz1. Diphthamide biosynthesis is essential for life because deficiency in any of the genes involved in its biosynthesis can cause embryo lethality during mouse development and mutations in human DPH1 gene can cause severe fatal diseases. In this manuscript, Shi et al identified compound heterozygous mutations of DPH1 gene (Q41X/E242Q) in a DEDSSH patient, and demonstrated that these DPH1 mutations caused the defect in diphthamide modification on eEF2. To model DEDSSH syndrome, the authors generated a mouse line with the compound mutations from the patient, and demonstrated that the mutant mice have the similar phenotypes of Dph1-KO mice in previous studies. Agreeing well with the previous studies, the authors demonstrated that Dph1 depletion decreased proliferation of several cell types. Thus far, the experiment were well performed and controlled and conclusions were supported by data.

However, the second half of the manuscript, the part claiming that the diphthamide-deficient eEF2 acts as a transitional co-activator, was weak, lacking convincing experimental data to support.

Major:

Figs. 4b, 4c: DPH1 knocked down cells still showed significant amount of DHP1 expression (approximately half by Q-PCR), which was likely to be sufficient for diphthamide modification (because Dph1^{+/-} mice were normal). Thus, upregulation of p21 and PUMA was unlikely caused by diphthamide deficiency.

Fig. 4e: Again, there was no evidence given to demonstrate p53-associated eEF2 was diphthamide-deficient.

The authors also need to analyze cells with diphthamide-deficiency due to KO (by CRISPR) other diphthamide genes (such as DPH2, DPH5, DPH6, and DPH7, which only function in diphthamide biosynthesis).

Fig. 5: p21^{-/-} mice are viable and fertile, thus it is unclear why the authors didn't use p21^{-/-} background for their rescue experiment?

Minor:

Fig. 1h: The eEF2 structure modeling suggest that the E242Q mutation in the proband may cause loss of the potential Glu242-Arg349 salt bridge, thereby affecting eEF2's stability. However, Fig. 1h did not support this because the eEF2 level in the proband cells remained the same as controls, even the mutant eEF2 was produced by only one allele. So may the structure distorted in some way caused the eEF2's function compromised.

Reviewer #2 (Remarks to the Author):

The manuscript 'Diphthamide-deficient eEF2 acts as a transcriptional co-activator for p53 to induce p21 expression and neural crest defects' by Shi et al., describes the recessive mutations of DPH1 in a DEDSSH patient with multiple defects in neural crest (NC) derivatives. To address the clinical findings, the authors generated the knock-in mice harboring the patient's mutations and Xenopus with Dph1 ablation. Mechanistically, the authors find that DPH1 depletion causes nonmodified eEF2 redistribution from ribosome to nucleus and interacts with p53 to transcriptionally induce p21 expression that possibly explain the proliferation reduction in NC-derived tissues. The study focusing on the role of diphthamide modification of eEF2 due to the newly identified recessive mutations of DPH1 in a patient with developmental delay and intellectual disability has clinical impact. However, the statement and interpretations of the results are confusing, and several concerns should be addressed.

1. The patient carried two mutated alleles (136C>T nonsense mutation and 724G>C missense mutation) of the DPH1 gene, which have not been reported previously. These mutant alleles presumably encoded DPH1(Q46X) and DPH1(E242Q) in lymphoblast cells of the patient. In the Fig. 1j, If the DPH1(Q46X) can be generated, it should be a truncated DPH1 protein and could not be detected at ~50-kDa protein size by the anti-DPH1 antibody. Thus, the paternal 724G>C mutant allele encoding DPH1(E242Q) was expressed and detected in the lane 2 of Fig. 1j. The DPH1(E242Q) levels in the patient was about 20-25% decreased compared to the DPH1 protein levels in the lymphoblast cells of unaffected sister or either parents (Fig. 1j). Based on this data, the authors speculate that the Glu242Gln mutation affects DPH1 stability (page 6). However, the author did not perform any experiment to show DPH1(E242Q) stability. This data could also be explained by the compensatory effect of paternal 724G>C mutant allele, which encoded DPH1(E242Q) and increased its expression up to about 80% of DPH1 levels. Therefore, the authors should conduct further experiments for the data interpretation to address whether E242Q mutation really affects on DPH1 stability.
2. Loss of function of DPH1 results in diphthamide-deficient eEF2. In the Fig. 1h and 1i, the authors examined the levels of diphthamide modified eEF2 and found that the diphthamide-eEF2 was reduced (~50%) in the patient's lymphoblasts compared to that in unaffected sister or either parents. Because the total eEF2 was unchanged among the samples, thus the ratio of diphthamide-eEF2/non-diphthamide-eEF2 in the patient's cells might be expected to be 1:1. This finding suggests that E242Q mutation of DPH1 partially impairs DPH1 function for diphthamide modification of eEF2. The authors based on structure modeling suggests that Glu242 forms a salt bridge with active site Arg349, which plays critical role for ACP transfer from SAM. Thus, the authors hypothesized that 'the E242Q mutation in the patient can be attributed to the loss of Glu242-Arg349 salt bridge' (page 7). However, it is unclear whether Glu242 mutation on the structure basis causes complete or incomplete diphthamide deficiency of eEF2. Does this E242Q mutation of DPH1 cause partial (or incomplete) diphthamide deficiency of eEF2 based on Figure 1h and Figure 1i? This should be explained in the result section.
3. The authors further created 'knockin (KI) mice harboring g either c.121C>T, p. Q41X or c.709G>C, p. E237Q mutant allele, which corresponds to human Q46X and E242Q mutations, respectively'(page 7). In the Extended Data Table S1, the progenies at embryonic stages ranging from E9.5-E16 were generated from intercrossing of Dph1(Q46X) and Dph1(E242Q) mice. The Table S1 is confusing, because it is difficult to understand which stage is critical for DPH1(Q46X)/E242Q compound mutants' growth or lethality. Also, the genotype labeling was confusing in Table S1 (labeled as Q46X/E242Q) and in main text (labeled as E237Q/Q41X; page 7).
4. In page 7, the authors described that 'Both the Dph1Q41X/+ and Dph1E237Q/+ single KI mice were born at expected Mendelian ratio (Extended Data Table S1) and appeared grossly normal, whereas the Dph1 E237Q/Q41X compound heterozygous KI mice died at E15.5 with markedly reduced body size (Fig. 2a-c).' However, the embryos at E10.5, but not E15.5, were showed in Fig. 2a-c. I recommend to show phenotypic analysis at both E10.5 and E15.5 stages. At E15.5, proliferation of NC-derived tissues (such plate shelves, cartilage of the skull, melanocytes etc) in control and Dph1 E237Q/Q41X KI embryos should be examined.
5. In the Extended Data Fig. S4, the authors explored other known DEDSSH-associated missense DPH1 mutations on DPH1 activity for diphthamide modification of eEF2. 'As compared with wild-type DPH1, all tested mutants showed reduced abilities to rescue eEF2 diphthamide modification when overexpressed in DPH1 KO HEK293T cells (Extended Data Fig. S4b-f).' Quantification of eEF2 diphthamide modification levels (or activity) of the tested DPH1 mutants compared with wild-type DPH1 in DPH1 KO HEK293T cells would be required to provide the fundamental insights in eEF2 diphthamide deficiency of these tested mutants compared with E242Q mutation (as shown in Fig. 1h and i).
6. Using Xenopus model for studying NC development, the authors find that dph1 KO inhibits cellular proliferation of neuroepithelium and reduces pre-migratory NC. While Xenopus model is a better system than mouse model for study NC proliferation and migration, the DPH1 critical mutations (E242Q and Q46X) have not been addressed. In fact, the NC-derived tissues (cranial skeletons, palate shelves, peripheral nerves etc) in Dph1 E237Q/Q41X compound heterozygous KI mouse embryos at E14.5-15.5 should be analyzed to provide the insights of DPH1 E242Q mutation on mammalian NC hypoplasia.
7. In Figure 4, the authors switched to U251 cell-based system to knockout or knockdown DPH1,

and then analyzed the interaction of non-diphthamide eEF2 and p53. PUMA and p21 appeared to be the p53 target genes, which could be enhanced in knockout/knockdown DPH1 in U251 cells as well as in Dph1 E237Q/Q41X embryos at E10.5. While PUMA was enhanced in the Dph1 mutant embryos, no apoptotic cells or markers were detected (Fig. S9). In contrast, DPH1 KO in U251 cells can inhibit cellular proliferation. However, it is unclear whether Dph1 E237Q/Q41X embryonic tissues, especially in NC derived tissues, also show decreased cellular proliferation; this is an important issue that should be addressed. Can elevated p21 protein be detected in Dph1 E237Q/Q41X embryos?

8. In Figures 4, S11 and S12, the authors demonstrated eEF2 nuclear translocation and interaction with p53. In Figure S11, the eEF2 band was blurry and fuzzy in nuclear and ribosomal fractions. Do the 'ribosomes' indicate polysomes in Figure S11? or total ribosomes, including free ribosomes? Does eEF2 not only interact with polysomes, but also with free ribosomes? Quantification of nuclear and ribosomal eEF2 normalized with total eEF2 should be presented. In Figure S12, the fluorescence images are also fuzzy and small. It is difficult to identify nuclear eEF2. In fact, eEF2 was mainly located in the cytosol based on the imaging data in Fig S12. In DPH1 KO cells, all eEF2 proteins are non-diphthamide eEF2. If the ratio of nuclear/cytosolic eEF2 is very low in DPH1 KO cells, what would be the ratio of nuclear/cytosolic eEF2 in DPH1 E242Q mutant cells? According to the data presented in Fig. 1h and 1i, DPH1 E242Q mutant cells may contain both diphthamide and non-diphthamide modified eEF2, contributing to even lower ratio of nuclear/cytosolic eEF2. It is a puzzle how does a small fraction of non-diphthamide eEF2, which can act as a cofactor of p53 and specifically target on some p53 target genes?

9. In Table S2, parental or maternal genotypes were mislabeled. How can DphQ41X,p21 mice cross with DphQ41X mice to generate Dph1 E237Q/Q41X embryos? The authors showed that loss of one allele of p21 appeared to partially rescue the reduced body size in Dph1E237Q/Q41X mutants (Fig. 5a). In Fig. 5b-e, histologic analyses of palate shaves were shown. Is this partial rescuing of growth and cleft palate due to an increase of cellular proliferation in NC of Dph1 E237Q/Q41X,p21+/- embryos? Can p21 null rescue Dph1 E237Q/Q41X embryonic lethal phenotype?

10. In Fig. 5f, the schematic model was presented. Under the DPH1 mutant conditions, no diphthamide was modified in eEF2 and resulted in nuclear localization and p53 binding for transcriptional activation of p21. However, the ratio of cytoplasmic/nuclear eEF2 without diphthamide is unclear. Apparently, the amount of cytosolic non-diphthamide eEF2 is higher than that in nuclei as a transcriptional co-activator, according to Fig. S12. Thus, the hypothetical model in Fig. 5f seems to be over-interpretation.

Typo: Title of Table S3; Materials and Methods- line 139: '...', with the primers shown in table S1' (Table S1 is the mouse breeding records); Gene nomenclature: human DPH1 and mouse Dph1 should be carefully distinguished.

Reviewer #3 (Remarks to the Author):

Shi et al found the new DEDSSH patient with compound heterozygous dph1 mutations. They generated knock-in mice corresponds to human E242Q and Q46X dph1 mutations. dph1[E237Q/Q41X] compound heterozygous mice died at E15.5. Lethal point was later in the previous dph1 KO mice, suggesting E237Q is the partial loss of function allele. They confirmed the function of this allele in cell-lysate based diphthamide modification assay.

Inhibition of proliferation was known to be a function of dph1 in some cell types. In this study, they showed that the mechanism is the nuclear translocation of unmodified eEF2 and its binding to p53, resulting in enhancement of p53 function. This is an important new findings for the diphthamide modification of eEF2. However, the basic mechanism of why the nuclear translocation of unmodified eEF2 and its binding to p53 results in p53 activation and why only some of the targets of p53 are transcriptionally upregulated remains to be analyzed. Specific comments below.

Craniofacial abnormality has been observed in patients with dph1 mutations, knock-in mice for the mutant allele generated in this study, and dph1 knock-out mice. Therefore, it is possible that dph1 mutation causes abnormalities in cells derived from the neural crest (NC). They analyzed dph1

function using *Xenopus* and found that dph1 MO attenuated the expression of NC markers (*pax3*, *msx1*, *zic1*) on the injected side. They also used embryos expressing the NC marker, *snai2:GFP*, and showed a CRISPER/Cas9-based F0 mosaic dph1 KO decreased GFP and pH3 signal. The decrease in pH3 by dph1 KO was not restored by eEF2 expression, but was restored by diphthamide-modified eEF2 mimic mutant (G717R) expression. No apoptosis was detected by TUNEL assay. Therefore, they concluded that diphthamide-modified eEF2 is required for proliferation of neuroepithelium containing NCs.

1. Apoptotic cells are quickly phagocytosed and eliminated by surrounding cells in embryo. Therefore, it is possible that they simply failed to detect apoptosis in this experiment. Contribution of apoptosis for *snai2:GFP* signal reduction by dph1 KO can be tested by overexpression of anti-apoptotic *bcl-2*.

2. In *Xenopus* experiment, effects of dph1 knockout are seen across the entire region where NC markers are expressed. However, craniofacial abnormality in patients with dph1 mutations or the dph1 mutant mice suggest specific region of NCs must be affected. Is this a species difference? It will be nice to follow the expression of NC markers (and pH3 staining) at each developmental stage in the dph1 mutant mice generated in this study.

3. Apoptosis has also been studied in mice using cleaved caspase-3 antibody (Fig. S9a-d), but immunochemical analysis of whole brain with different stages will be required. Apoptosis induction may also occur in dph1 mutant mice, since the expression of proapoptotic PUMA, a proapoptotic BH3 protein, is induced in dph1 KD U251 cells.

The inhibition of neuroepithelium proliferation led them to consider p53 involvement. eEF2 is not a ribosomal protein, but craniofacial abnormality has been observed in some ribosomopathies and involvement of p53 for ribosomopathies is reported (Farley-Barnes et al., TIG 35, 754, 2019). Also, in a previous dph1 KO study (Chen et al., G&D 18, 320, 2004), dph1KO MEFs showed growth inhibition, which was restored by p53 mutation. Thus, it is predicted that dph1 and p53 are cooperatively involved in cell proliferation. In the present study, they examined the interaction between eEF2 and p53 more directly. dph1 KO in U251 causes growth inhibition. Neither mRNA nor protein expression of p53 was altered during this process, but there was an increase in expression of p53 targets (PUMA and p21). Therefore, they considered that dph1 may be involved in p53 function rather than p53 expression and stability. co-IP experiment showed eEF2-p53 binding. More binding was observed with dph1 siRNA.

4. Is this mainly due to the localization change of unmodified eEF2 to the nucleus, or the change of binding affinity of unmodified eEF2 to p53? It is possible to examine whether the binding of unmodified eEF2 or modified eEF2 to p53 is direct or not by using GST-pulldown assay. GST-eEF2 can be expressed and purified from *E. coli* and yeast for this assay.

5. In contrast to U251, dph1 KD does not change localization of eEF2 in HEK293. Does co-IP of eEF2 and p53 change in HEK293 with or without Dph1 siRNA?

6. In Fig. S11, it appears that not much transfer of eEF2 to the nucleus has occurred but there is a decrease of eEF2 in the ribosomal fraction. Is this reproducible? If so, how does this occur? In Fig. S12, quality of the immunohistochemistry to show the eEF2 localization needs to be improved and quantitative analysis is required.

7. ChIP-qPCR showed that binding of p53 to the promoter region of p21 and PUMA increased in dph1 KD. This supports that mRNA increases in the same condition. However, how eEF2-p53 binding increases transcription is totally unknown. At least, eEF2 binding motif to p53 needs to be determined, and experiments using eEF2 mutants that fail to bind to p53 are required.

Reviewer #1:

Diphthamide is a modified Histidine residue on eEF2, with its modification depends on a protein complex including Dph1 to Dph7, as well as a regulator Miz1. Diphthamide biosynthesis is essential for life because deficiency in any of the genes involved in its biosynthesis can cause embryo lethality during mouse development and mutations in human DPH1 gene can cause severe fatal diseases. In this manuscript, Shi et al identified compound heterozygous mutations of DPH1 gene (Q41X/E242Q) in a DEDSSH patient, and demonstrated that these DPH1 mutations caused the defect in diphthamide modification on eEF2. To model DEDSSH syndrome, the authors generated a mouse line with the compound mutations from the patient, and demonstrated that the mutant mice have the similar phenotypes of Dph1-KO mice in previous studies. Agreeing well with the previous studies, the authors demonstrated that Dph1 depletion decreased proliferation of several cell types. Thus far, the experiment were well performed and controlled and conclusions were supported by data. However, the second half of the manuscript, the part claiming that the diphthamide-deficient eEF2 acts as a transitional co-activator, was weak, lacking convincing experimental data to support.

Major:

Figs. 4b, 4c: DPH1 knocked down cells still showed significant amount of DHP1 expression (approximately half by Q-PCR), which was likely to be sufficient for diphthamide modification (because Dph1^{+/-} mice were normal). Thus, upregulation of p21 and PUMA was unlikely caused by diphthamide deficiency.

Response: Thank the reviewer's comment. The lymphoblastoid cells derived from the proband's mother, who had only one functional allele of *DPH1* (Fig. 1g), had merely <15% and statistically insignificant reduction in DPH1 protein level (Fig. 1j), as well as insignificant reduction in eEF2 diphthamide modification (Fig. 1i). Similar to the *DPH1*^{+/-} mice, the proband's mother also appeared to be healthy (see Extended Data Patient Information). These suggest that there is a feedback to boost the expression of the remaining functional *DPH1* allele in the cells to compensate for the loss of one allele, which may explain the lack of phenotypes in *DPH1*^{+/-} mice (as there is little reduction in DPH1 expression as compared with wild-type control). In contrast, we were able to achieve a statistically significant, 50-60% reduction in DPH1 protein level in U251 cells with two separate siRNAs (Fig. 4b). This reduction is more severe than in the proband-derived lymphoblastoid cells, where there was a ~23% reduction in DPH1 protein that led to >50% reduction in diphthamide-modified eEF2 (Fig. 1i, j). It is therefore not surprising that DPH1 knockdown can lead to p21 and PUMA upregulation while the *DPH1*^{+/-} mice are grossly normal. We apologize for the lack of clarity, and have revised the text to reflect the possible feedback regulation of DPH1 levels (highlighted in yellow in pp. 6).

The upregulation of p21 mRNA and protein has been demonstrated in **a)** *DPH1* knockout U251 cells generated using CRISPR/Cas9 (Fig. 4a); **b)** U251 cells with DPH1 knockdown using two separate siRNAs (Fig. 4b, c); **c)** lysates of *Dph1*^{E237Q/Q41X} mouse embryos (Fig. 4d, e); and **d)** U251 cells with knockdown of DPH2 and DPH4, which are also required for diphthamide biosynthesis (this revision; see below). While PUMA is not the focus of the current manuscript, we also found it to be upregulated in U251 cells with DPH1 knockdown using either siRNA and *Dph1*^{E237Q/Q41X} mouse embryo lysates (Fig. 4b, c, e). Thus, we are

confident that the upregulation of p21 and PUMA is specific for diphthamide deficiency and not an off-target effect.

Fig. 4e: Again, there was no evidence given to demonstrate p53-associated eEF2 was diphthamide-deficient. The authors also need to analyze cells with diphthamide-deficiency due to KO (by CRISPR) other diphthamide genes (such as DPH2, DPH5, DPH6, and DPH7, which only function in diphthamide biosynthesis).

Response: We thank the reviewer for this thoughtful suggestion, and we have carried out siRNA-mediated knockdown of DPH2 and DPH6 in U251 cells. RT-qPCR controls show ~80% and 90% reduction of *DPH2* and *DPH6* mRNA upon knockdown, and in both cases we detected a significant upregulation of p21 (Fig. S11 of the current version; 3 biological replicates each). These new results confirm that p21 upregulation is a general consequence of diphthamide deficiency (see highlighted text in pp. 11-12).

It is unlikely that only diphthamide-deficient eEF2 is associated with p53, as we were able to detect a significant amount of p53 pulled down by an anti-eEF2 antibody in control U251 cells without DPH1 knockdown (Fig. 4f, g), which contain little diphthamide-deficient eEF2 (note that nearly 100% of eEF2 is diphthamide modified in mammalian cells; Tsuda-Sakurai and Miura, *J. Biochem.* 2019, 165: 1-8; Liu et al., *PNAS* 2012, 109:13817-13822). However, DPH1 knockdown greatly enhanced the amount of p53 pulled down using the anti-eEF2 antibody, and reduced ribosome-associated eEF2 (Fig. 4g, h). In the structure of eEF2 associated with ribosomes, diphthamide is localized at the tip of domain IV and interacts with rRNA (Pellegrino et al., *J. Mol. Biol.* 2018, 430:2677-2687). Therefore, a straightforward model is that p53 competes with ribosomes for eEF2 like a tug of war, and that diphthamide deficiency reduces the affinity of eEF2 for ribosomes, shifting the balance toward p53 and resulting in more p21 transcription. We apologize for the lack of clarity in the original version of this manuscript, and have revised the title, abstract and the introduction (highlighted text in pp. 1, 3 and 5) to avoid this confusion. A clearer explanation of our model has also been added to the discussion (highlighted text in pp. 13).

Fig. 5: p21^{-/-} mice are viable and fertile, thus it is unclear why the authors didn't use p21^{-/-} background for their rescue experiment?

Response: We did not use *p21*^{-/-} background for the rescue because we were able to achieve a significant rescue using the *p21*^{+/-} background, which is easier to obtain. Moreover, p21 is well known to have anti-apoptotic function (Gartel and Tyner, *Mol. Cancer Ther.* 2002, 1:639-649; Kreis et al., *Cancers* 2019, 11091220). In particular, it has been shown that the choice between growth arrest and apoptosis upon p53 pathway activation is dependent on the balance between p21 and PUMA, as a complete knockout of p21 leads to robust PUMA-mediated apoptosis (Polyak et al., *Genes Dev.* 1996, 10:1945-1952; Yu et al., *PNAS* 2003, 100:1931-1936). We found elevated *Puma* expression but did not detect any apparent increase in apoptosis in *Dph1*^{E237Q/Q41X} mouse embryos (Fig. 4e and S9), possibly due to a concomitant upregulation of p21. Consequently, it is possible that using the *p21*^{-/-} background for the rescue experiments may cause PUMA-induced apoptosis and a failure of rescue.

Minor:

Fig. 1h: The eEF2 structure modeling suggest that the E242Q mutation in the proband may cause loss of the potential Glu242-Arg349 salt bridge, thereby affecting eEF2's stability. However, Fig. 1h did not support this because the eEF2 level in the proband cells remained the same as controls, even the mutant eEF2 was produced by only one allele. So may the structure distorted in some way caused the eEF2's function compromised.

Response: The mutations occurred in *DPH1*, not *eEF2*, and the structural modeling was also for the *DPH1* protein. However, the reviewer is correct that the structure of *DPH1* may be distorted in some way only causing compromised function but not stability. We have made changes in the text correspondingly (see our response to Reviewer #2, Point 1).

Reviewer #2:

The manuscript 'Diphthamide-deficient eEF2 acts as a transcriptional co-activator for p53 to induce p21 expression and neural crest defects' by Shi et al., describes the recessive mutations of *DPH1* in a DEDSSH patient with multiple defects in neural crest (NC) derivatives. To address the clinical findings, the authors generated the knock-in mice harboring the patient's mutations and *Xenopus* with *Dph1* ablation. Mechanistically, the authors find that *DPH1* depletion causes nonmodified eEF2 redistribution from ribosome to nucleus and interacts with p53 to transcriptionally induce p21 expression that possibly explain the proliferation reduction in NC-derived tissues. The study focusing on the role of diphthamide modification of eEF2 due to the newly identified recessive mutations of *DPH1* in a patient with developmental delay and intellectual disability has clinical impact. However, the statement and interpretations of the results are confusing, and several concerns should be addressed.

1. The patient carried two mutated alleles (136C>T nonsense mutation and 724G>C missense mutation) of the *DPH1* gene, which have not been reported previously. These mutant alleles presumably encoded *DPH1*(Q46X) and *DPH1*(E242Q) in lymphoblast cells of the patient. In the Fig. 1j, If the *DPH1*(Q46X) can be generated, it should be a truncated *DPH1* protein and could not be detected at ~50-kDa protein size by the anti-*DPH1* antibody. Thus, the paternal 724G>C mutant allele encoding *DPH1*(E242Q) was expressed and detected in the lane 2 of Fig. 1j. The *DPH1*(E242Q) levels in the patient was about 20-25% decreased compared to the *DPH1* protein levels in the lymphoblast cells of unaffected sister or either parents (Fig. 1j). Based on this data, the authors speculate that the Glu242Gln mutation affects *DPH1* stability (page 6). However, the author did not perform any experiment to show *DPH1*(E242Q) stability. This data could also be explained by the compensatory effect of paternal 724G>C mutant allele, which encoded *DPH1*(E242Q) and increased its expression up to about 80% of *DPH1* levels. Therefore, the authors should conduct further experiments for the data interpretation to address whether E242Q mutation really affects on *DPH1* stability.

Response: The reviewer had a great point here. Our initial data suggested that the E242Q mutation caused a reduction in the *DPH1* protein level, when the lymphoblastoid cells from the proband's father (*DPH1*^{+/E242Q}) were compared with those from her sister (*DPH1*^{+/+}). However, after repeating the experiment several more times and performing the statistical analyses carefully, we found that the differences between the father and the sister and between the proband (*DPH1*^{E242Q/Q46X}) and the mother (*DPH1*^{+/Q46X}) are not significant,

although there is still a trend of reduction in both cases (Fig. 1j). As the reviewer pointed out, there is a compensatory upregulation of the *DPH1*^{E242Q} allele that further complicates the analysis. Therefore, we have softened the tone on the effect of E242Q mutation on DPH1 stability and added discussion on the compensatory expression of the non-truncated allele (highlighted in pp. 6-7). Since the Glu242 residue is involved in forming a salt bridge with the Arg349 residue that is critical for binding SAM and forming the ACP-histidine intermediate (Fig. 1l and S1), it is possible that the E242Q mutation, which leads to the loss of Glu242-Arg349 salt bridge, interferes with the local positioning of SAM and impairs ACP transfer. Further testing this hypothesis requires solving the structures of the wild-type DPH1 and the E242 mutant, which is beyond the scope of this manuscript. We feel that while this is an important question, it does not affect the main conclusion of this manuscript, which is the induction of p21 expression and neural crest defects by diphthamide deficiency, and hope the reviewer agrees with us.

2. Loss of function of DPH1 results in diphthamide-deficient eEF2. In the Fig. 1h and 1i, the authors examined the levels of diphthamide modified eEF2 and found that the diphthamide-eEF2 was reduced (~50%) in the patient's lymphoblasts compared to that in unaffected sister or either parents. Because the total eEF2 was unchanged among the samples, thus the ratio of diphthamide-eEF2/non-diphthamide-eEF2 in the patient's cells might be expected to be 1:1. This finding suggests that E242Q mutation of DPH1 partially impairs DPH1 function for diphthamide modification of eEF2. The authors based on structure modeling suggests that Glu242 forms a salt bridge with active site Arg349, which plays critical role for ACP transfer from SAM. Thus, the authors hypothesized that 'the E242Q mutation in the patient can be attributed to the loss of Glu242-Arg349 salt bridge' (page 7). However, it is unclear whether Glu242 mutation on the structure basis causes complete or incomplete diphthamide deficiency of eEF2. Does this E242Q mutation of DPH1 cause partial (or incomplete) diphthamide deficiency of eEF2 based on Figure 1h and Figure 1i? This should be explained in the result section.

Response: As the reviewer stated, the proband-derived lymphoblastoid cells (*DPH1*^{E242Q/Q46X}) retained nearly 50% of diphthamide-modified eEF2, as compared with the sister-derived cells (*DPH1*^{+/+}; Fig. 1i). Since the Q46X mutations likely results in a complete loss of DPH1 function, the remaining diphthamide biosynthesis activity should be solely from the E242Q mutant. The amount of DPH1 protein produced by this mutant was close to 80% of the amount generated by the two wild-type alleles in the sister-derived lymphoblastoid cells (Fig. 1j). Thus, we believe that the E242Q mutation causes partial diphthamide deficiency. An explanation has been added to pp. 7, Lines 10-12 (highlighted in yellow), as suggested by the reviewer.

3. The authors further created 'knockin (KI) mice harboring g either c.121C>T, p. Q41X or c.709G>C, p. E237Q mutant allele, which corresponds to human Q46X and E242Q mutations, respectively' (page 7). In the Extended Data Table S1, the progenies at embryonic stages ranging from E9.5-E16 were generated from intercrossing of *Dph1*(Q46X) and *Dph1*(E242Q) mice. The Table S1 is confusing, because it is difficult to understand which stage is critical for *DPH1*(Q46X)/E242Q compound mutants' growth or lethality. Also, the genotype labeling was confusing in Table S1 (labeled as Q46X/E242Q) and in main text (labeled as E237Q/Q41X; page 7).

Response: We apologize for the mistakes and thank the reviewer for pointing this out. An updated Table S1 with the corrections has been uploaded. The growth defect was apparent as early as E10.5 (Fig. 3a). As for the timing of lethality, we have obtained live *Dph1*^{E237Q/Q41X} embryos as late as E16.5 but never any newborns. Therefore, we have rephrased it as “...all *Dph1*^{E237Q/Q41X} compound heterozygous KI mice died before birth...” (highlighted in pp. 9-10).

4. In page 7, the authors described that ‘Both the *Dph1*Q41X/+ and *Dph1*E237Q/+ single KI mice were born at expected Mendelian ratio (Extended Data Table S1) and appeared grossly normal, whereas the *Dph1* E237Q/Q41X compound heterozygous KI mice died at E15.5 with markedly reduced body size (Fig. 2a-c).’ However, the embryos at E10.5, but not E15.5, were showed in Fig. 2a-c. I recommend to show phenotypic analysis at both E10.5 and E15.5 stages. At E15.5, proliferation of NC-derived tissues (such plate shelves, cartilage of the skull, melanocytes etc) in control and *Dph1* E237Q/Q41X KI embryos should be examined.

Response: As the reviewer suggested, we analyzed the phenotypes at E10.5 and E16.5 (as some embryos survived to E16.5, although most died by E15.5). The *Dph1*^{E237Q/Q41X} embryos displayed reduced pH3 and the NC marker SOX10 at E10.5, as well as smaller mandible and shortened Meckel’s cartilage at E16.5 (see new Fig. 3d-g, p-s and highlighted text in pp. 10, Lines 10-12). These phenotypes are consistent with NC hypoplasia observed in *Xenopus* embryos (Fig. 2d-g).

5. In the Extended Data Fig. S4, the authors explored other known DEDSSH-associated missense DPH1 mutations on DPH1 activity for diphthamide modification of eEF2. ‘As compared with wild-type DPH1, all tested mutants showed reduced abilities to rescue eEF2 diphthamide modification when overexpressed in DPH1 KO HEK293T cells (Extended Data Fig. S4b-f).’ Quantification of eEF2 diphthamide modification levels (or activity) of the tested DPH1 mutants compared with wild-type DPH1 in DPH1 KO HEK293T cells would be required to provide the fundamental insights in eEF2 diphthamide deficiency of these tested mutants compared with E242Q mutation (as shown in Fig. 1h and i).

Response: We completely agree with the reviewer, and have repeated these experiments and added quantification and statistics as suggested. As compared with wild-type DPH1, each of the disease-associated mutants (m1-m5) had significantly less rescue of the diphthamide-modified eEF2, when expressed in DPH1 KO HEK293 cells (Fig. S3g, h of the current version).

6. Using *Xenopus* model for studying NC development, the authors find that *dph1* KO inhibits cellular proliferation of neuroepithelium and reduces pre-migratory NC. While *Xenopus* model is a better system than mouse model for study NC proliferation and migration, the DPH1 critical mutations (E242Q and Q46X) have not been addressed. In fact, the NC-derived tissues (cranial skeletons, palate shelves, peripheral nerves etc) in *Dph1* E237Q/Q41X compound heterozygous KI mouse embryos at E14.5-15.5 should be analyzed to provide the insights of DPH1 E242Q mutation on mammalian NC hypoplasia.

Response: Please see our response to Point 4 above. Briefly, our new data suggest hypoplasia of the NC (at E10.5) and mandible (at E16.5) in *Dph1*^{E237Q/Q41X} embryos.

7. In Figure 4, the authors switched to U251 cell-based system to knockout or knockdown DPH1, and then analyzed the interaction of non-diphthamide eEF2 and p53. PUMA and p21 appeared to be the p53 target genes, which could be enhanced in knockout/knockdown DPH1 in U251 cells as well as in *Dph1* E237Q/Q41X embryos at E10.5. While PUMA was enhanced in the *Dph1* mutant embryos, no apoptotic cells or markers were detected (Fig. S9). In contrast, DPH1 KO in U251 cells can inhibit cellular proliferation. However, it is unclear whether *Dph1* E237Q/Q41X embryonic tissues, especially in NC derived tissues, also show decreased cellular proliferation; this is an important issue that should be addressed. Can elevated p21 protein be detected in *Dph1* E237Q/Q41X embryos?

Response: This is another great question. As mentioned above, we did observe decreased pH3 levels in the *Dph1*^{E237Q/Q41X} knockin embryos using IHC (new Fig. 3d, e). Our initial attempts of western blotting using mouse embryo lysates failed due to high background noise; therefore, only RT-qPCR results were shown in the original version of this manuscript. After troubleshooting, we were able to obtain clean blots for p21, histone H3 and phosphorylated H3 from mouse embryo lysates, and indeed detected an increase in p21 protein and decrease in H3 phosphorylation (new Fig. 4d).

8. In Figures 4, S11 and S12, the authors demonstrated eEF2 nuclear translocation and interaction with p53. In Figure S11, the eEF2 band was blur and fuzzy in nuclear and ribosomal fractions. Do the 'ribosomes' indicate polysomes in Figure S11? or total ribosomes, including free ribosome? Does eEF2 not only interact with polysomes, but also with free ribosomes? Quantification of nuclear and ribosomal eEF2 normalized with total eEF2 should be presented. In Figure S12, the fluorescence images are also fuzzy and small. It is difficult to identify nuclear eEF2. In fact, eEF2 was mainly located in the cytosol based on the imaging data in Fig S12. In DPH1 KO cells, all eEF2 proteins are non-diphthamide eEF2. If the ratio of nuclear/cytosolic eEF2 is very low in DPH1 KO cells, what would be the ratio of nuclear/cytosolic eEF2 in DPH1 E242Q mutant cells? According to the data presented in Fig. 1h and 1i, DPH1 E242Q mutant cells may contain both diphthamide and non-diphthamide modified eEF2, contributing to even lower ratio of nuclear/cytosolic eEF2. It is puzzle how does a small fraction of non-diphthamide eEF2, which can act as a cofactor of p53 and specifically target on some p53 target genes?

Response: We have repeated the experiments in the former Fig. S11 and carried out quantification and statistical analyses. Images of the new blots and quantification/statistics have been added as new Fig. 4h in the current version. Quantification and statistics of nuclear eEF2 in 200 lymphoblastoid cells of each genotype (*DPH1*^{+/+} and *DPH1*^{E242Q/Q46X}) have also been performed (new Fig. S13b). These results are consistent with our conclusion that diphthamide deficiency leads to decreased eEF2 association with ribosome and increased nuclear translocation.

The "ribosomes" here primarily refer to polysomes. We have tried to purify ribosomes by following the protocol described by Belin et al., *Curr. Protoc. Cell Biol.* 2010, 3:3.40 with high (4 M) and low (0.01M) concentrations of KCl, which are suitable for isolating monosomes

and polysomes, respectively. High KCl yielded little associated eEF2 (not shown), whereas low KCl gave clearly detectable signals in western blotting for eEF2 (Fig. 4h). Therefore, the translation elongation factor eEF2 is mainly associated with polysomes, consistent with the current consensus that the vast majority of peptide bonds are formed on polysomes, with monosomes mainly responsible for translating special types of open reading frames (Heyer and Moore, *Cell* 2016, 164:757-769).

As suggested by the reviewer, we quantified total, ribosomal and nuclear eEF2 in control and DPH1 KO cells, and the results are consistent with our previous observation that eEF2 is reduced in the ribosomes and increased in the nuclei (new Fig. 4h). We would like to point out that it is the absolute amount of unmodified eEF2, but not the ratio of unmodified vs. modified or nuclear vs. cytosolic eEF2, that is important for regulating p53. In mammalian cells, nearly 100% of eEF2 is diphthamide modified (Tsuda-Sakurai and Miura, *J. Biochem.* 2019, 165: 1-8; Liu et al., *PNAS* 2012, 109:13817-13822). Thus, a moderate diphthamide deficiency can result in a drastic upregulation of unmodified eEF2, which has higher binding affinity for p53 and enhances p21 transcription. It should also be noted that eEF2 is a highly abundant protein (as it is the component of every actively translating ribosome in the cells), whereas p53 levels are kept low via degradation mediated by MDM2 etc. in normal cells (Moll and Petrenko, *Mol. Cancer Res.* 2003, 1:1001-1008). Therefore, even a small fraction of unmodified eEF2 may be sufficient to bind a reasonably large fraction of p53, leading to significant increase in p21 and PUMA mRNA and protein.

9. In Table S2, parental or maternal genotypes were mislabeled. How can DphQ41X,p21 mice cross with DphQ41X mice to generate Dph1 E237Q/Q41X embryos? The authors showed that loss of one allele of p21 appeared to partially rescue the reduced body size in Dph1E237Q/Q41X mutants (Fig. 5a). In Fig. 5b-e, histologic analyses of palate shaves were showed. Is this partial rescuing of growth and cleft palate due to an increase of cellular proliferation in NC of Dph1 E237Q/Q41X,p21^{+/-} embryos? Can p21 null rescue Dph1 E237Q/Q41X embryonic lethal phenotype?

Response: We thank the reviewer for pointing out the mislabeling in Table S2, and have corrected these mistakes. The rescue of growth and cleft palate can be attributed to increased cell proliferation as a consequence of loss of one *p21* allele, but we were unable to test directly due to the short timeframe and large amount of work to be completed for the revision. We were also unable to carry out the rescue in p21-null background but suspect that a complete loss of p21 may induce apoptosis, resulting in a failure of rescue (see our response to Reviewer #1 in pp. 2, last paragraph of this rebuttal letter).

10. In Fig. 5f, the schematic model was presented. Under the DPH1 mutant conditions, no diphthamide was modified in eEF2 and resulted in nuclear localization and p53 binding for transcriptional activation of p21. However, the ratio of cytoplasmic/nuclear eEF2 without diphthamide is unclear. Apparently, the amount of cytosolic non-diphthamide eEF2 is higher than that in nuclei as a transcriptional co-activator, according to Fig. S12. Thus, the hypothetical model in Fig. 5f seems to be over-interpretation.

Response: We agree with the reviewer that the cytosolic portion of eEF2 is still higher than the nuclear portion in diphthamide deficiency, and have revised the model to reflect this. However, it is the absolute amount of unmodified eEF2, but not the ratio of cytoplasmic/nuclear eEF2, that is important for inducing p21 expression (see our response to Point 8 above).

Typo: Title of Table S3; Materials and Methods- line 139: ‘..., with the primers shown in table S1’ (Table S1 is the mouse breeding records); Gene nomenclature: human DPH1 and mouse Dph1 should be carefully distinguished.

Response: We apologize for the mistakes and have corrected them.

Reviewer #3:

Shi et al found the new DEDSSH patient with compound heterozygous dph1 mutations. They generated knock-in mice corresponds to human E242Q and Q46X dph1 mutations. dph1[E237Q/Q41X] compound heterozygous mice died at E15.5. Lethal point was later in the previous dph1 KO mice, suggesting E237Q is the partial loss of function allele. They confirmed the function of this allele in cell-lysate based diphthamide modification assay. Inhibition of proliferation was known to be a function of dph1 in some cell types. In this study, they showed that the mechanism is the nuclear translocation of unmodified eEF2 and its binding to p53, resulting in enhancement of p53 function. This is an important new findings for the diphthamide modification of eEF2. However, the basic mechanism of why the nuclear translocation of unmodified eEF2 and its binding to p53 results in p53 activation and why only some of the targets of p53 are transcriptionally upregulated remains to be analyzed.

Specific comments below.

Craniofacial abnormality has been observed in patients with dph1 mutations, knock-in mice for the mutant allele generated in this study, and dph1 knock-out mice. Therefore, it is possible that dph1 mutation causes abnormalities in cells derived from the neural crest (NC). They analyzed dph1 function using *Xenopus* and found that dph1 MO attenuated the expression of NC markers (pax3, msx1, zic1) on the injected side. They also used embryos expressing the NC marker, *snai2:GFP*, and showed a CRISPER/Cas9-based F0 mosaic dph1 KO decreased GFP and pH3 signal. The decrease in pH3 by dph1 KO was not restored by eEF2 expression, but was restored by diphthamide-modified eEF2 mimic mutant (G717R) expression. No apoptosis was detected by TUNEL assay. Therefore, they concluded that diphthamide-modified eEF2 is required for proliferation of neuroepithelium containing NCs.

1. Apoptotic cells are quickly phagocytosed and eliminated by surrounding cells in embryo. Therefore, it is possible that they simply failed to detect apoptosis in this experiment. Contribution of apoptosis for *snai2:GFP* signal reduction by dph1 KO can be tested by overexpression of anti-apoptotic *bcl-2*.

Response: We agree with the reviewer that we cannot completely rule out the contribution of apoptosis to the phenotypes. Since we did not detect enhanced apoptosis in either *Xenopus* embryos with DPH1 knockout or the *Dph1*^{E237Q/Q41X} double-knockin mouse embryos, and loss of one *p21* allele rescued the phenotypes in the double-knockin mice (Fig. 5), we believe

that the predominant effect caused by diphthamide deficiency is reduced proliferation, which has also been shown in various other models (reviewed in Tsuda-Sakurai and Miura, *J. Biochem.* 2019, 165: 1-8). In addition, ectopic expression of XR11, the *Xenopus* homolog of BCL2, alone can cause expansion of the NC and loss of NC boundary in *Xenopus* embryos (Tríbulo et al., *Dev. Biol.* 2004, 275:325-342), complicating the interpretation of the effects of *dph1* KO on eGFP expression in the *snai2:eGFP* transgenic frog embryos.

2. In *Xenopus* experiment, effects of *dph1* knockout are seen across the entire region where NC markers are expressed. However, craniofacial abnormality in patients with *dph1* mutations or the *dph1* mutant mice suggest specific region of NCs must be affected. Is this a species difference? It will be nice to follow the expression of NC markers (and pH3 staining) at each developmental stage in the *dph1* mutant mice generated in this study.

Response: Besides craniofacial disorders, the proband also had sparse hair and growth hormone deficiency (Extended Data Patient Information), which are typical NC defects. Therefore, the proband appeared to have general NC defects that affect multiple tissues, consistent with the broad effects of *dph1* knockout across the entire NC region in *Xenopus* embryos. It has been well established in the NC field that KO of genes required for early NC development in non-mammalian vertebrates (many of which are also required for NC induction from human pluripotent stem cells) do not typically cause similar phenotypes in mice; instead, late and tissue-specific NC development is often perturbed. The reason for this discrepancy remains unclear, but question on whether mouse is suitable for modeling early human NC development has been raised (Barriga et al., *Development* 2015, 142: 1555-1560). To avoid any species bias, we used both *Xenopus* and mouse as models in this study. We did carry out IHC for pH3 and the NC marker SOX10 in E10.5 *Dph1*^{E237Q/Q41X} mouse embryos, as the reviewer suggested, and found a reduction of both. While this stage is relatively late for NC development (post-migratory), the phenotypes are still consistent with our conclusion that diphthamide deficiency leads to reduced NC proliferation.

3. Apoptosis has also been studied in mice using cleaved caspase-3 antibody (Fig. S9a-d), but immunochemical analysis of whole brain with different stages will be required. Apoptosis induction may also occur in *dph1* mutant mice, since the expression of proapoptotic PUMA, a proapoptotic BH3 protein, is induced in *dph1* KD U251 cells. The inhibition of neuroepithelium proliferation led them to consider p53 involvement. eEF2 is not a ribosomal protein, but craniofacial abnormality has been observed in some ribosomopathies and involvement of p53 for ribosomopathies is reported (Farley-Barnes et al., *TIG* 35, 754, 2019). Also, in a previous *dph1* KO study (Chen et al., *G&D* 18, 320, 2004), *dph1*KO MEFs showed growth inhibition, which was restored by p53 mutation. Thus, it is predicted that *dph1* and p53 are cooperatively involved in cell proliferation. In the present study, they examined the interaction between eEF2 and p53 more directly. *dph1* KO in U251 causes growth inhibition. Neither mRNA nor protein expression of p53 was altered during this process, but there was an increase in expression of p53 targets (PUMA and p21). Therefore, they considered that *dph1* may be involved in p53 function rather than p53 expression and stability. co-IP experiment showed eEF2-p53 binding. More binding was observed with *dph1* siRNA.

Response: The absence of apparent apoptosis in cells with diphthamide deficiency, despite an upregulation of PUMA, may be due to the concomitant upregulation of p21, as p21 is well known to have anti-apoptotic function and can counter the pro-apoptotic activity of PUMA (see our response to Reviewer #1 in pp. 2, last paragraph of this rebuttal letter). Since we were able to obtain a significant rescue of the major phenotypes (growth and craniofacial) in *Dph1*^{E237Q/Q41X} double-knockin mouse embryos by knocking out one allele of *p21* (Fig. 5), and the effects of diphthamide deficiency on proliferation has been well documented (reviewed in Tsuda-Sakurai and Miura, *J. Biochem.* 2019, 165: 1-8), we believe that the primary pathway that leads to the major phenotypes caused by diphthamide deficiency is p21-mediated growth arrest, although we cannot rule out additional contributions of apoptosis.

4. Is this mainly due to the localization change of unmodified eEF2 to the nucleus, or the change of binding affinity of unmodified eEF2 to p53? It is possible to examine whether the binding of unmodified eEF2 or modified eEF2 to p53 is direct or not by using GST-pulldown assay. GST-eEF2 can be expressed and purified from *E. coli* and yeast for this assay.

Response: We have started to investigate the detailed mechanisms of eEF2-p53 interaction, and found that it is the N-terminal GTPase domains (residues 1-486) that are necessary and sufficient for binding p53 (see our response to Point 7 below). Since the diphthamide-modified His715 is outside of this region, it is unlikely that diphthamide modification directly regulates p53 binding. In the structure of eEF2 associated with ribosomes, diphthamide is localized at the tip of domain IV and interacts with mRNA and rRNA (Pellegrino et al., *J. Mol. Biol.* 2018, 430:2677-2687). Therefore, a straightforward model is that p53 competes with ribosomes for eEF2 like a tug of war, and diphthamide deficiency shifts the balance toward p53 by reducing the affinity of eEF2 for ribosomes, resulting in more eEF2-bound p53 with enhanced association with the *p21* promoter. A brief discussion on this has been added to pp. 13 (highlighted in yellow). It is also possible that diphthamide modification affects the global conformation of eEF2 (as suggested by Liu et al., *PNAS* 2012, 109:13817-13822), resulting in altered accessibility of the N-terminal GTPase domains to p53 or the C-terminal nuclear localization signals. A complete understanding of these mechanisms will require tremendous time and effort, and is beyond the scope of this study – we hope the reviewer agrees with us.

5. In contrast to U251, *dph1* KD does not change localization of eEF2 in HEK293. Does co-IP of eEF2 and p53 change in HEK293 with or without *Dph1* siRNA?

Response: The original low-passage *dph1* KO HEK293T cells generated using CRISPR-Cas9 had very low proliferation rate, consistent with other cells with diphthamide deficiency. However, we noticed that after a few passages, these cells started to pick up their growth speed and became indistinguishable from the parental HEK293T cells, possibly due to suppressor mutations that gave the cells growth advantage. This may explain the lack of an apparent effect on eEF2 localization in our previous results. Since regenerating these KO cells are very time-consuming, we decided to take out these data and instead focus on U251 and lymphoblastoid cells (see our response to Point 6 below).

6. In Fig. S11, it appears that not much transfer of eEF2 to the nucleus has occurred but there is a decrease of eEF2 in the ribosomal fraction. Is this reproducible? If so, how does this occur? In Fig. S12, quality of the immunohistochemistry to show the eEF2 localization needs to be improved and quantitative analysis is required.

Response: We have repeated the experiments shown in Fig. S11 of the original version of this manuscript, and found that both the decrease of eEF2 in the ribosomal fraction and increase in the nuclear fraction are significant. These data have now been added as Fig. 4h. We have also quantified the eEF2 nuclear localization shown in former Fig. S12 (now Fig. S13) and added statistics to show that it is significantly increased in patient-derived (*Dph1*^{E242Q/Q46X}) lymphoblastoid cells.

7. ChIP-qPCR showed that binding of p53 to the promoter region of p21 and PUMA increased in dph1 KD. This supports that mRNA increases in the same condition. However, how eEF2-p53 binding increases transcription is totally unknown. At least, eEF2 binding motif to p53 needs to be determined, and experiments using eEF2 mutants that fail to bind to p53 are required.

Response: As the reviewer suggested, we generated truncated mutants of eEF2. The mutant containing the N-terminal GTPase domains (eEF2¹⁻⁴⁸⁶) retains the binding for p53, whereas the mutant encompassing the C-terminal tRNA-mimicking domains (eEF2⁴⁸⁷⁻⁸⁵⁸) fails to bind p53 (new Fig. S12 and pp. 12, Lines 7-13). This is an interesting observation, as p53 has been shown to interact with small GTPases such as Rac1 (Yue et al., *Genes & Dev.* 2017, 31:1641-1654). How this interaction affects *p21* transcription remains unclear, but it has been shown that p53-mediated transcription of distinct target genes, such as *p21*, can be selectively regulated by various factors including specific residues and post-translational modifications on p53, as well as epigenetic events and non-coding RNAs (see newly added discussion in pp. 14, highlighted in yellow).

Reviewer #1 (Remarks to the Author):

The authors have made good efforts to address the reviewers' concerns. However, I still have the following concerns:

1. p21^{+/-} rescue experiments were poorly presented. The basis for the rescue was assumed the lower levels of p21 mRNA and protein (~50% of WT) in p21^{+/-} mice. However, the authors did not demonstrate this is true. Were there some Dph1E237Q/Q41X;p21^{+/-} mice born live?
2. Because p21^{-/-} mice are viable, I still do not understand why p21^{-/-} mice were not used for the rescue experiments. The authors explained that using p21^{-/-} background for the rescue experiments MAY cause PUMA-induced apoptosis. But this needs to be shown if this is true. The authors claimed that phenotypes of the mutant mice and thus the patient are caused by increased association of eEF2 (diphthamide-deficient) with p53, it is even more logical to use p53^{-/-} background (also viable) to reuse Dph1 mutant mice.
3. There is potential caveat for the co-IP experiments using anti-eEF2 antibody. Several anti-eEF2 antibodies can recognize/bind diphthamide-deficient eEF2 much better than the WT eEF2 at native condition (but equally well at denatured condition) (ref 1). Thus, anti-eEF2 may pull down more p53/eEF2 (diphthamide-deficient) than p53/eEF2 (WT). To rule out this, the authors need to probe both eEF2 and p53 in Western blotting for the co-IP products (at denatured condition) (Fig. 4g).

Reviewer #2 (Remarks to the Author):

This revised manuscript incorporates supplemental data to comprehensively address and clarify my previous queries while rectifying the inconsistencies found in the prior version. Therefore, I have no additional comments to make regarding this revision.

Reviewer #3 (Remarks to the Author):

In this revised MS, the authors have answered the issues I raised for the original version with additional experiments. It still remains unclear why unmodified eEF2 is more likely to leave the ribosome and how unmodified eEF2 regulates p53 function. However, this study is important because it shows that the phenotype of dph1 mutation is caused by a p21-mediated failure of NC cell proliferation.

Reviewer #1:

The authors have made good efforts to address the reviewers' concerns. However, I still have the following concerns:

1. $p21^{+/-}$ rescue experiments were poorly presented. The basis for the rescue was assumed the lower levels of p21 mRNA and protein (~50% of WT) in $p21^{+/-}$ mice. However, the authors did not demonstrate this is true. Were there some $Dph1^{E237Q/Q41X};p21^{+/-}$ mice born live?

Response: As the reviewer suggested, we carried out Western blotting for p21 in wild-type, $p21^{+/-}$ and $p21^{-/-}$ mice. Indeed, there was ~50% reduction of the p21 protein in $p21^{+/-}$ mice as compared with their wild-type littermates (new Fig. S14c and d; highlighted text in pp. 13, 2nd paragraph).

We have not obtained any $Dph1^{E237Q/Q41X} p21^{+/-}$ mice born live, but the sample size was not large enough to draw any conclusion on whether loss of one $p21$ allele can rescue the death phenotype in $Dph1^{E237Q/Q41X}$ mice. We would also like to point out that the phenotypes we focused on in this manuscript, such as the craniofacial defects and smaller body size that are likely the consequences of elevated p21 in $Dph1^{E237Q/Q41X}$ mice, do not usually cause death before birth. Instead, our phenotypic and transcriptomic studies revealed an additional defect in definitive hematopoiesis in our $Dph1^{E237Q/Q41X}$ mice (not shown), which has also been observed in $Dph1^{-/-}$ mice (Ref. 9). Since mice lacking definitive hematopoiesis typically die around E15 (about the same time most of our $Dph1^{E237Q/Q41X}$ mice die; Yamane T., *Front. Cell Dev. Biol.* 2018), we suspect that the hematopoiesis defect is the actual cause of death of these mice. We do not know if this hematopoiesis defect is p21-dependent at this point; hence we cannot predict if reducing $p21$ can rescue this defect and/or the death phenotype in $Dph1^{E237Q/Q41X}$ mice. While we are actively pursuing this hematopoiesis defect, we believe that a complete understanding of the underpinning mechanisms will take quite some time and is beyond the scope of the current study.

2. Because $p21^{-/-}$ mice are viable, I still do not understand why $p21^{-/-}$ mice were not used for the rescue experiments. The authors explained that using $p21^{-/-}$ background for the rescue experiments MAY cause PUMA-induced apoptosis. But this needs to be shown if this is true. The authors claimed that phenotypes of the mutant mice and thus the patient are caused by increased association of eEF2 (diphthamide-deficient) with p53, it is even more logical to use $p53^{-/-}$ background (also viable) to reuse $Dph1$ mutant mice.

Response: We thank the reviewer for this legitimate criticism, but would like to point out respectfully the reasons why we did not use $p21^{-/-}$ mice for the rescue experiments. First, mice with the $Dph1^{E237Q/Q41X};p21^{-/-}$ genotype are rare and especially difficult to obtain when the appropriate littermate controls (wild-type, $Dph1^{E237Q/Q41X}$, etc.) are needed at the same time, as phenotypes such as body size can have large variations from litter to litter. Second, since we have already observed a significant rescue of both the body size and cleft palate phenotypes in $Dph1^{E237Q/Q41X}$ mice by deleting one $p21$ allele (Fig. 5a-e), we are not sure if repeating the same rescue experiments with both $p21$ alleles deleted could provide much more mechanistic insight. Last, as we explained in the previous rebuttal letter, a complete loss of p21 is known to permit the induction of PUMA-mediated apoptosis (Polyak et al., *Genes Dev.* 1996, 10:1945-1952; Yu et al., *PNAS* 2003, 100:1931-1936), which may complicate

data analyses. Indeed, our TUNEL staining results (see figure below) suggest a marked upregulation of apoptosis in the craniofacial structures of *p21*^{-/-} mice as compared to wild-type littermates, which was further increased in *Dph1*^{E237Q/Q41X};*p21*^{-/-} mice. In contrast, *Dph1*^{E237Q/Q41X} mice did not show elevated apoptosis, although they had enhanced *Puma* expression (Fig. 4e, Fig. S9 and the figure below). These observations are consistent with our hypothesis that apoptosis in *Dph1*^{E237Q/Q41X} mice is suppressed by p21 but becomes apparent when p21 is completely lost.

Figure. TUNEL staining of palatal shelf sections from E15.5 mice with the indicated genotypes. TUNEL signals are shown in green, and nuclei were stained with DAPI. WT, wild-type.

We did not pursue a rescue with the *p53*^{-/-} background because a) we had difficulties acquiring the *p53* knockout mice, b) our *Dph1*^{E237Q/Q41X} mice had enhanced expression of p21 but not p53 (Fig. 4d, e), and c) Chen and Behringer have already shown that both the reduced body size and cleft palate in *Dph1*^{-/-} mice can be rescued by loss of p53 (Ref. 9). Thus, we believe that showing a successful rescue by reducing the dosage of p21 would be more insightful, and hope the reviewer agrees with us.

3. There is potential caveat for the co-IP experiments using anti-eEF2 antibody. Several anti-eEF2 antibodies can recognize/bind diphthamide-deficient eEF2 much better than the WT eEF2 at native condition (but equally well at denatured condition) (ref 1). Thus, anti-eEF2 may pull down more p53/eEF2 (diphthamide-deficient) than p53/eEF2 (WT). To rule out this, the authors need to probe both eEF2 and p53 in Western blotting for the co-IP products (at denatured condition) (Fig. 4g).

Response: The reviewer raised a great point here. We have now added this important control, which shows that our anti-eEF2 antibody pulled down the same amount of eEF2, as detected by western blotting under denatured condition (IP: eEF2, WB: eEF2 in Fig. 4g).

Reviewer #2:

This revised manuscript incorporates supplemental data to comprehensively address and clarify my previous queries while rectifying the inconsistencies found in the prior version. Therefore, I have no additional comments to make regarding this revision.

Response: We thank the reviewer for the positive opinion towards this manuscript.

Reviewer #3:

In this revised MS, the authors have answered the issues I raised for the original version with additional experiments. It still remains unclear why unmodified eEF2 is more likely to leave the ribosome and how unmodified eEF2 regulates p53 function. However, this study is important because it shows that the phenotype of dph1 mutation is caused by a p21-mediated failure of NC cell proliferation.

Response: We would like to thank the reviewer for being enthusiastic about this study. In fact, our data do not indicate that unmodified eEF2 is more likely to leave the ribosome; they only suggest that unmodified eEF2 has increased association with p53 and decreased association with the ribosome, as compared with diphthamide-modified eEF2. These alterations could be caused by enhanced binding affinity of unmodified eEF2 for p53, or reduced affinity for the ribosome, or both. We apologize for the confusion, and have revised the *Introduction* (highlighted in pp.5, 2nd paragraph) and *Discussion* (highlighted in pp.13, last paragraph) to clarify this point. The detailed mechanisms underlying these alterations, as well as how unmodified eEF2 regulates p53 function, are still under investigation and are beyond the scope of this study. Again, we thank the reviewer for the understanding and support.

Reviewer #1 (Remarks to the Author):

The authors have addressed my concerns in this version. Thus I do not have further comments.